# Exponential Bellman Equation and Improved Regret Bounds for Risk-Sensitive Reinforcement Learning

**Yingjie Fei**[1]   **Zhuoran Yang**[2]   **Yudong Chen**[3]   **Zhaoran Wang**[1]
[1] Northwestern University; `yf275@cornell.edu, zhaoranwang@gmail.com`
[2] Princeton University; `zy6@princeton.edu`
[3] University of Wisconsin-Madison; `yudong.chen@wisc.edu`

## Abstract

We study risk-sensitive reinforcement learning (RL) based on the entropic risk measure. Although existing works have established non-asymptotic regret guarantees for this problem, they leave open an exponential gap between the upper and lower bounds. We identify the deficiencies in existing algorithms and their analysis that result in such a gap. To remedy these deficiencies, we investigate a simple transformation of the risk-sensitive Bellman equations, which we call the exponential Bellman equation. The exponential Bellman equation inspires us to develop a novel analysis of Bellman backup procedures in risk-sensitive RL algorithms, and further motivates the design of a novel exploration mechanism. We show that these analytic and algorithmic innovations together lead to improved regret upper bounds over existing ones.

## 1   Introduction

Risk-sensitive reinforcement learning (RL) is important for practical and high-stake applications, such as self-driving and robotic surgery. In contrast with standard and risk-neutral RL, it optimizes some risk measure of cumulative rewards instead of their expectation. One foundational framework for risk-sensitive RL maximizes the entropic risk measure of the reward, which takes the form of

$$V^\pi = \frac{1}{\beta} \log\{\mathbb{E}_\pi[e^{\beta R}]\},$$

with respect to the policy $\pi$, where $\beta \neq 0$ is a given risk parameter and $R$ denotes the cumulative rewards.

Recently, the works of [20, 21] investigate the online setting of the above risk-sensitive RL problem. Under $K$-episode MDPs with horizon length of $H$, they propose two model-free algorithms, namely RSVI and RSQ, and prove that their algorithms achieve the regret upper bound (with its informal form given by)

$$\text{Regret}(K) \lesssim e^{|\beta|H^2} \cdot \frac{e^{|\beta|H} - 1}{|\beta|H} \sqrt{\text{poly}(H) \cdot K}$$

without assuming knowledge of the transition distribution or access to a simulator. They also provide a lower bound (informally presented as)

$$\text{Regret}(K) \gtrsim \frac{e^{|\beta|H'} - 1}{|\beta|H} \sqrt{\text{poly}(H) \cdot K}$$

that any algorithm has to incur, where $H'$ is a linear function of $H$. Despite the non-asymptotic nature of their results, it is not hard to see that a wide gap exists between the two bounds. Specifically,

35th Conference on Neural Information Processing Systems (NeurIPS 2021).

the upper bound has an additional $e^{|\beta|H^2}$ factor compared to the lower bound, and even worse, this factor is dominating in the upper bound since the quadratic exponent in $e^{|\beta|H^2}$ makes it exponentially larger than $\frac{e^{|\beta|H}-1}{|\beta|H}$ even for moderate values of $|\beta|$ and $H$. It is unclear whether the factor of $e^{|\beta|H^2}$ is intrinsic in the upper bound.

In this paper, we show that the additional factor in the upper bound is not intrinsic for the upper bound and can be eliminated by a refined algorithmic design and analysis. We identify two deficiencies in the existing algorithms and their analysis: (1) the main element of the analysis follows existing analysis of risk-neutral RL algorithms, which fails to exploit the special structure of the Bellman equations of risk-sensitive RL; (2) the existing algorithms use an excessively large bonus that results in the exponential blow-up in the regret upper bound.

To address the above shortcomings, we consider a simple transformation of the Bellman equations analyzed so far in the literature, which we call the *exponential Bellman equation*. A distinctive feature of the exponential Bellman equation is that they associate the instantaneous reward and value function of the next step in a multiplicative way, rather than in an additive way as in the standard Bellman equations. From the exponential Bellman equation, we develop a novel analysis of the Bellman backup procedure for risk-sensitive RL algorithms that are based on the principle of optimism. The analysis further motivates a novel exploration mechanism called *doubly decaying* bonus, which helps the algorithms adapt to their estimation error over each horizon step while at the same time exploring efficiently. These discoveries enable us to propose two model-free algorithms for RL with the entropic risk measure based on the novel bonus. By combining the new analysis and bonus design, we prove that the preceding algorithms attain nearly optimal regret bounds under episodic and finite-horizon MDPs. Compared to prior results, our regret bounds feature an exponential improvement with respect to the horizon length and risk parameter, removing the factor of $e^{|\beta|H^2}$ from existing upper bounds. This significantly narrows the gap between upper bounds and the existing lower bound of regret.

In summary, we make the following theoretical contributions in this paper.

1. We investigate the gap between existing upper and lower regret bounds in the context of risk-sensitive RL, and identify deficiencies of the existing algorithms and analysis;

2. We consider the exponential Bellman equation, which inspires us to propose a novel analysis of the Bellman backup procedure for RL algorithms based on the entropic risk measure. It further motivates a novel bonus design called doubly decaying bonus. We then design two model-free risk-sensitive RL algorithms equipped with the novel bonus.

3. The novel analytic framework and bonus design together enable us to prove that the preceding algorithms achieve nearly optimal regret bounds, which improve upon existing ones by an exponential factor in terms of the horizon length and risk sensitivity.

## 2   Related works

The problem of RL with respect to the entropic risk measure is first proposed by the classical work of [24], and has since inspired a large body of studies [2, 4–8, 13, 16–18, 22, 23, 25, 26, 31, 33, 37, 38, 40, 41, 43]. However, the algorithms from this line of works require knowledge of the transition kernel or assume access to a simulator of the underlying environment. Theoretical properties of these algorithms are investigated based on these assumptions, but the results are mostly of asymptotic nature, which do not shed light on their dependency on key parameters of the environment and agent.

The work of [20] represents the first effort to investigate the setting where transitions are unknown and simulators of the environment are unavailable. It establishes the first non-asymptotic regret or sample complexity guarantees under the tabular setting. Building upon [20], the authors of [21] extend the results to the function approximation setting, by considering linear and general function approximations of the underlying MDPs. Nevertheless, as discussed in Section 1, both works leave open an exponential gap between the regret upper and lower bounds, which the present work aims to address via novel algorithms and analysis motivated by the exponential Bellman equation.

We remark that although the exponential Bellman equation has been previously investigated in the literature of risk-sensitive RL [2, 5], this is the first time that it is explored for deriving regret and sample complexity guarantees of risk-sensitive RL algorithms. In Appendix A, we also make

connections between risk-sensitive RL and distributional RL through the exponential Bellman equation.

**Notations.** For a positive integer $n$, we let $[n] := \{1, 2, \ldots, n\}$. For two non-negative sequences $\{a_i\}$ and $\{b_i\}$, we write $a_i \lesssim b_i$ if there exists a universal constant $C > 0$ such that $a_i \leq Cb_i$ for all $i$, and write $a_i \asymp b_i$ if $a_i \lesssim b_i$ and $b_i \lesssim a_i$. We use $\tilde{O}(\cdot)$ to denote $O(\cdot)$ while hiding logarithmic factors. For functions $f, g : \mathcal{U} \to \mathbb{R}$, where $\mathcal{U}$ denotes their domain, we write $f \geq g$ if $f(u) \geq g(u)$ for any $u \in \mathcal{U}$. We denote by $\mathbb{I}\{\cdot\}$ the indicator function.

# 3 Problem background

## 3.1 Episodic and finite-horizon MDP

The setting of episodic Markov decision processes can be denoted by $\mathrm{MDP}(\mathcal{S}, \mathcal{A}, H, \mathcal{P}, \mathcal{R})$, where $\mathcal{S}$ is the set of states, $\mathcal{A}$ is the set of actions, $H \in \mathbb{Z}_{>0}$ is the length of each episode, and $\mathcal{P} = \{P_h\}_{h \in [H]}$ and $\mathcal{R} = \{r_h\}_{h \in [H]}$ are the sets of transition kernels and reward functions, respectively. We let $S := |\mathcal{S}|$ and $A := |\mathcal{A}|$, and we assume $S, A < \infty$. We let $P_h(\cdot \mid s, a)$ denote the probability distribution over successor states of step $h + 1$ if action $a$ is executed in state $s$ at step $h$. We assume that the reward function $r_h : \mathcal{S} \times \mathcal{A} \to [0, 1]$ is deterministic. We also assume that both $\mathcal{P}$ and $\mathcal{R}$ are unknown to learning agents.

Under the setting of an episodic MDP, the agent aims to learn the optimal policy by interacting with the environment throughout $K > 0$ episodes, described as follows. At the beginning of episode $k$, an initial state $s_1^k$ is selected by the environment and we assume $s_1^k$ stays the same for all $k \in [K]$. In each step $h \in [H]$ of episode $k$, the agent observes state $s_h^k \in \mathcal{S}$, executes an action $a_h^k \in \mathcal{A}$, and receives a reward equal to $r_h(s_h^k, a_h^k)$ from the environment. The MDP then transitions into state $s_{h+1}^k$ randomly drawn from the transition kernel $P_h(\cdot \mid s_h^k, a_h^k)$. The episode terminates at step $H + 1$, in which the agent does not take actions or receive rewards. We define a policy $\pi = \{\pi_h\}_{h \in [H]}$ as a collection of functions $\pi_h : \mathcal{S} \to \mathcal{A}$, where $\pi_h(s)$ is the action that the agent takes in state $s$ at step $h$ of the episode.

## 3.2 Risk-sensitive RL

For each $h \in [H]$, we define the value function $V_h^\pi : \mathcal{S} \to \mathbb{R}$ of a policy $\pi$ as the cumulative utility of the agent at state $s$ of step $h$ under the entropic risk measure, assuming that the agent commits to policy $\pi$ in later steps. Specifically, we define

$$\forall (h, s) \in [H] \times \mathcal{S}, \quad V_h^\pi(s) := \frac{1}{\beta} \log \left\{ \mathbb{E}\left[ e^{\beta \sum_{i=h}^H r_i(s_i, \pi_i(s_i))} \,\Big|\, s_h = s \right] \right\}, \tag{1}$$

where $\beta \neq 0$ is a given risk parameter. The agent aims to maximize his cumulative utility in step 1, that is, to find a policy $\pi$ such that $V_1^\pi(s)$ is maximized for all state $s \in \mathcal{S}$. Under this setting, if $\beta > 0$, the agent is risk-seeking and if $\beta < 0$, the agent is risk-averse. Furthermore, as $\beta \to 0$ the agent tends to be risk-neutral and $V_h^\pi(s)$ tends to the classical value function.

We may also define the action-value function $Q_h^\pi : \mathcal{S} \times \mathcal{A} \to \mathbb{R}$, which is the cumulative utility of the agent who follows policy $\pi$, conditional on a particular state-action pair; formally, this is given by

$$\forall (h, s, a) \in [H] \times \mathcal{S} \times \mathcal{A}, \quad Q_h^\pi(s, a) := \frac{1}{\beta} \log \left\{ \mathbb{E}\left[ e^{\beta \sum_{i=h}^H r_i(s_i, a_i)} \,\Big|\, s_h = s, a_h = a \right] \right\}, \tag{2}$$

Under some mild regularity conditions [2], there always exists an optimal policy, which we denote as $\pi^*$, that yields the optimal value $V_h^*(s) := \sup_\pi V_h^\pi(s)$ for all $(h, s) \in [H] \times \mathcal{S}$.

**Bellman equations.** For all $(s, a) \in \mathcal{S} \times \mathcal{A}$, the Bellman equation associated with a policy $\pi$ is given by

$$Q_h^\pi(s, a) = r_h(s, a) + \frac{1}{\beta} \log \left\{ \mathbb{E}_{s' \sim P_h(\cdot \mid s, a)} \left[ e^{\beta \cdot V_{h+1}^\pi(s')} \right] \right\}, \tag{3}$$

$$V_h^\pi(s) = Q_h^\pi(s, \pi(s)), \qquad V_{H+1}^\pi(s) = 0$$

for $h \in [H]$. In Equation (3), it can be seen that the action value $Q_h^\pi$ of step $h$ is a non-linear function of the value function $V_{h+1}^\pi$ of the later step. This is in contrast with the linear Bellman equations in the risk-neutral setting ($\beta \to 0$), where $Q_h^\pi(s,a) = r_h(s,a) + \mathbb{E}_{s'}[V_{h+1}^\pi(s')]$. Based on Equation (3), for $h \in [H]$, the Bellman optimality equation is given by

$$Q_h^*(s,a) = r_h(s,a) + \frac{1}{\beta} \log \left\{ \mathbb{E}_{s' \sim P_h(\cdot \,|\, s,a)} \left[ e^{\beta \cdot V_{h+1}^*(s')} \right] \right\}, \tag{4}$$

$$V_h^*(s) = \max_{a \in \mathcal{A}} Q_h^*(s,a), \qquad V_{H+1}^*(s) = 0.$$

**Exponential Bellman equation.** We introduce the *exponential Bellman equation*, which is an exponential transformation of Equations (3) and (4) (by taking exponential on both sides): for any policy $\pi$ and tuple $(h,s,a)$, we have

$$e^{\beta \cdot Q_h^\pi(s,a)} = \mathbb{E}_{s' \sim P_h(\cdot \,|\, s,a)} \left[ e^{\beta(r_h(s,a) + V_{h+1}^\pi(s'))} \right]. \tag{5}$$

When $\pi = \pi^*$, we obtain the corresponding optimality equation

$$e^{\beta \cdot Q_h^*(s,a)} = \mathbb{E}_{s' \sim P_h(\cdot \,|\, s,a)} \left[ e^{\beta(r_h(s,a) + V_{h+1}^*(s'))} \right]. \tag{6}$$

Note that Equation (5) associates the current and future cumulative utilities ($Q_h^\pi$ and $V_{h+1}^\pi$) in a multiplicative way. An implication of Equation (5) is that one may estimate $e^{\beta \cdot Q_h^\pi(s,a)}$ by a quantity of the form

$$w_h(s,a) = \text{SampAvg}(\{ e^{\beta(r_h(s_h,a_h) + V_{h+1}(s_{h+1}))} : (s_h, a_h) = (s,a) \}) \tag{7}$$

given some estimate of the value function $V_{h+1}$. Here, we denote by $\text{SampAvg}(\mathcal{X})$ the sample average computed over elements in the set $\mathcal{X}$ throughout past episodes, and it can be seen as an empirical MGF of cumulative rewards from step $h + 1$. Equation (5) also suggests the following policy improvement procedure for a risk-sensitive policy $\pi$:

$$\pi_h(s) \leftarrow \underset{a' \in \mathcal{A}}{\arg\max}\, Q_h(s,a') = \begin{cases} \arg\max_{a' \in \mathcal{A}} e^{\beta \cdot Q_h(s,a')}, & \text{if } \beta > 0 \\ \arg\min_{a' \in \mathcal{A}} e^{\beta \cdot Q_h(s,a')}, & \text{if } \beta < 0, \end{cases} \tag{8}$$

where $Q_h$ denotes some estimated action-value function, possibly obtained from the quantity $w_h$.

In the next section, we will discuss how the exponential Bellman equation (5) inspires the development of a novel analytic framework for risk-sensitive RL. Before proceeding, we introduce a performance metric for the agent. For each episode $k$, recall that $s_1^k$ is the initial state chosen by the environment and let $\pi^k$ be the policy of the agent at the beginning of episode $k$. Then the difference $V_1^*(s_1^k) - V_1^{\pi^k}(s_1^k)$ is called the *regret* of the agent in episode $k$. Therefore, after $K$ episodes, the total regret for the agent is given by

$$\text{Regret}(K) \coloneqq \sum_{k \in [K]} [V_1^*(s_1^k) - V_1^{\pi^k}(s_1^k)], \tag{9}$$

which serves as the key performance metric studied in this paper.

## 4 Analysis of risk-sensitive RL

### 4.1 Mechanism of existing analysis

In this section, we provide an informal overview of the mechanism underlying the existing analysis of risk-sensitive RL. Let us focus on the case $\beta > 0$ for simplicity of exposition; similar reasoning holds for $\beta < 0$. A key step in the existing regret analysis of RL algorithms is to establish a recursion on the difference $V_h^k - V_h^{\pi^k}$ over $h \in [H]$, where $V_h^k$ is the iterate of an algorithm in step $h$ of episode $k$ and $V_h^{\pi^k}$ is the value function of the policy used in episode $k$. Such approach can be commonly found in the literature of algorithms that use the upper confidence bound [27, 28], in which the recursion takes the form of

$$V_h^k - V_h^{\pi^k} \le V_{h+1}^k - V_{h+1}^{\pi^k} + \psi_h^k, \tag{10}$$

for $\beta \to 0$ and some quantity $\psi_h^k$. The work of [20], which studies the risk-sensitive setting under the entropic risk measure, also follows this approach and derives regret bounds by establishing the recursion of the form

$$V_h^k - V_h^{\pi^k} \le e^{\beta H}\left(V_{h+1}^k - V_{h+1}^{\pi^k}\right) + \frac{1}{\beta}\tilde{b}_h^k + e^{\beta H}\tilde{m}_h^k, \tag{11}$$

where $\tilde{b}_h^k$ denotes the bonus which enforces the upper confidence bound and leads to the inequality $V_h^k \ge V_h^\pi$ for any policy $\pi$, and $\tilde{m}_h^k$ is part of a martingale difference sequence. The derivation of Equation (11) is based on the Bellman equation (3), which shows that the action value $Q_h^{\pi^k}$ is the sum of the reward $r_h$ and the entropic risk measure of $V_{h+1}^{\pi^k}$. Following [20], we may then unroll the recursion (11) from $h = H$ to $h = 1$ to get

$$V_1^k - V_1^{\pi^k} \le \frac{1}{\beta}e^{\beta H^2}\sum_h \tilde{b}_h^k + e^{\beta H^2}\sum_h \tilde{m}_h^k, \tag{12}$$

given that $V_{H+1}^k = V_{H+1}^{\pi^k} = 0$. Using the inequality $\mathrm{Regret}(K) \le \sum_k(V_1^k - V_1^{\pi^k})$, $\sum_{k,h}\tilde{b}_h^k \lesssim (e^{\beta H} - 1)\sqrt{K}$ and $\sum_{k,h}\tilde{m}_h^k \lesssim \sqrt{K}$, we obtain the regret bound in [20] as $\mathrm{Regret}(K) \lesssim e^{\beta H^2}\frac{e^{\beta H}-1}{\beta H}\sqrt{K}$. Therefore, it can be seen that the dominating factor $e^{\beta H^2}$ in their regret bound originates in Equation (12), which can be further traced back to the exponential factor $e^{\beta H}$ in the error dynamics (11).

## 4.2 Refined approach via exponential Bellman equation

While the existing analysis in (11) is motivated by the Bellman equation of the form given in (3), we propose to work on the exponential Bellman equation (5). Equation (5) operates on the quantities $e^{\beta \cdot Q_h^\pi}$ and $e^{\beta \cdot V_{h+1}^\pi}$, which can be thought of as the MGFs of the current and future values, while the reward function $r_h$ is involved as a multiplicative term. This motivates us to derive a new recursion:

$$e^{\beta \cdot V_h^k} - e^{\beta \cdot V_h^{\pi^k}} \le e^{\beta \cdot r_h^k}\left(e^{\beta \cdot V_{h+1}^k} - e^{\beta \cdot V_{h+1}^{\pi^k}}\right) + b_h^k + m_h^k, \tag{13}$$

where $b_h^k, m_h^k$ denote some bonus and martingale terms, respectively, and $r_h^k$ stands for the reward in step $h$ of episode $k$. Unrolling Equation (13) yields

$$e^{\beta \cdot V_1^k} - e^{\beta \cdot V_1^{\pi^k}} \le \sum_h e^{\beta \cdot D_h^k}(b_h^k + m_h^k), \tag{14}$$

where $D_h^k = \sum_{i \in [h-1]} r_i^k$. In words, the error of $e^{\beta \cdot V_1^k} - e^{\beta \cdot V_1^{\pi^k}}$ is bounded by the weighted sum of bonus and martingale difference terms, where the weights are given by $e^{\beta \cdot D_h^k}$, the exponential rewards up to step $h - 1$. We may then apply a localized linearization of the logarithmic function, which gives $\mathrm{Regret}(K) \le \frac{1}{\beta}\sum_k(e^{\beta \cdot V_1^k} - e^{\beta \cdot V_1^{\pi^k}})$, and arrives at a regret upper bound (the formal regret bounds will be established in Theorems 1 and 2 below). Different from Equation (11) where rewards are only implicitly encoded in $V_h^k$, in Equation (13) rewards are explicitly involved in the error dynamics via an exponential term.

To see why Equation (13) is intuitively correct, we may divide both sides of the equation by $\beta$ and take $\beta \to 0$. By doing so, we should expect to obtain quantities from the error dynamics (10) of risk-neutral RL. Since the function $f_\beta(x) = (e^{\beta x} - 1)/\beta$ satisfies that $f_\beta(x) \to x$ as $\beta \to 0$ for any fixed $x$, we have

$$\lim_{\beta \to 0}\frac{1}{\beta}\left(e^{\beta \cdot V_h^k} - e^{\beta \cdot V_h^{\pi^k}}\right) = V_h^k - V_h^{\pi^k},$$

$$\lim_{\beta \to 0}\frac{1}{\beta}\left(e^{\beta \cdot r_h^k}\left(e^{\beta \cdot V_{h+1}^k} - e^{\beta \cdot V_{h+1}^{\pi^k}}\right)\right) = r_h^k + V_{h+1}^k - (r_h^k + V_{h+1}^{\pi^k}) = V_{h+1}^k - V_{h+1}^{\pi^k},$$

recovering terms in (10). Therefore, the recursion (13) can be seen as generalizing those in the analysis of risk-neutral RL.

By comparing Equations (13) and (11), we see that while both error dynamics are derived from the same underlying Bellman equation, they inspire drastically different forms of recursion. Note that

---

**Algorithm 1** RSVI2

---

1: $Q_h(\cdot, \cdot), V_h(\cdot) \leftarrow H - h + 1$, $N_h(\cdot, \cdot) \leftarrow 0$ and $w_h(\cdot, \cdot) \leftarrow 0$ for all $h \in [H + 1]$
2: **for** episode $k = 1, \ldots, K$ **do**
3:     **for** step $h = H, \ldots, 1$ **do**
4:         **for** $(s, a) \in \mathcal{S} \times \mathcal{A}$ such that $N_h(s, a) \geq 1$ **do**
5:              $w_h(s, a) \leftarrow \frac{1}{N_h(s,a)} \sum_{\tau \in [k-1]} \mathbb{I}\{(s_h^\tau, a_h^\tau) = (s, a)\} \cdot e^{\beta[r_h(s,a) + V_{h+1}(s_{h+1}^\tau)]}$
6:              $b_h(s, a) \leftarrow c|e^{\beta(H-h+1)} - 1|\sqrt{\frac{S \log(HSAK/\delta)}{N_h(s,a)}}$ where $c > 0$ is a universal constant
7:              $G_h(s, a) \leftarrow \begin{cases} \min\{w_h(s, a) + b_h(s, a), \ e^{\beta(H-h+1)}\}, & \text{if } \beta > 0 \\ \max\{w_h(s, a) - b_h(s, a), \ e^{\beta(H-h+1)}\}, & \text{if } \beta < 0 \end{cases}$
8:              $V_h(s) \leftarrow \max_{a' \in \mathcal{A}} \frac{1}{\beta} \log\{G_h(s, a')\}$
9:         **end for**
10:     **end for**
11:     $\forall h \in [H]$, take $a_h \leftarrow \text{argmax}_{a' \in \mathcal{A}} \frac{1}{\beta} \log\{G_h(s_h, a')\}$; observe $r_h(s_h, a_h), s_{h+1}$
12:     Add 1 to $N_h(s_h, a_h)$
13: **end for**

---

the multiplicative factor $e^{\beta \cdot r_h^k}$ in Equation (13) is milder than the factor $e^{\beta H}$ in Equation (11), since $r_h^k \in [0, 1]$. This is the source of an improvement of our refined analysis over existing works. On the other hand, the success of applying the error dynamics (13) in our analysis crucially depends on the choice of bonus terms $\{b_h^k\}$, as an improper choice would blow up the error $e^{\beta \cdot V_1^k} - e^{\beta \cdot V_1^{\pi^k}}$. This observation motivates our novel bonus design, as we explain next in Section 5.

## 5 Algorithms

### 5.1 Overview of algorithms

In this section, we propose two model-free algorithms for RL with the entropic risk measure. We first present RSVI2, which is based on value iteration, in Algorithm 1. The algorithm has two main stages: it first estimates the value function using data accumulated up to episode $k - 1$ (Line 3–10) and then executes the estimated policy to collect new trajectory (Line 11). In value function estimation, it computes the weights $w_h$, or the empirical MGF of some estimated cumulative rewards evaluated at $\beta$, which can be seen as a simple moving average over $\tau \in [k - 1]$. Therefore, Line 5 functions as a concrete implementation of Equation (7) where the sample average is instantiated as a simple moving average. Then in Line 7, it computes an augmented estimate $G_h$ by combining $w_h$ with a bonus term $b_h$ (defined in Line 6). This is followed by thresholding to put $G_h$ in the proper range. Note that $G_h$ is an optimistic estimator of the quantity $e^{\beta \cdot Q_h^\pi}$ in Equation (5): the construction of $G_h$ is augmented by $b_h$ so that it encourages exploration of rarely visited state-action pairs in future episodes, and thereby follows the principle of Risk-Sensitive Optimism in the Face of Uncertainty [20]. When $\beta < 0$, the bonus is subtracted from $w_h$, since a higher level of optimism corresponds to a *smaller* value of the estimate. In addition, Line 11 follows the reasoning of policy improvement suggested in Equation (8).

Next, we introduce RSQ2 in Algorithm 2, which is based on Q-learning. Similar to Algorithm 1, it consists of value estimation (Line 8–11) and policy execution (Line 6) steps. By combining Lines 9 and 10, we see that Algorithm 2 computes the optimistic estimate $G_h$ as a projection of an exponential moving average of empirical MGFs:

$$G_h(s_h, a_h) \leftarrow \Pi_h\{\text{EMA}(\{e^{\beta[r_h(s_h, a_h) + V_{h+1}(s_{h+1})]}\})\}, \tag{15}$$

where $\Pi_h$ denotes a projection that depends on step $h$. In particular, Line 9 can be interpreted as a computation of empirical MGFs evaluated at $\beta$ and thus a concrete implementation of Equation (7) using an exponential moving average. This is in contrast with the simple moving average update in Algorithm 1.

Although Algorithms 1 and 2 are inspired by RSVI and RSQ of [20], respectively, we note that the main novelty of our algorithms lies in the bonus terms ($b_h$ in Algorithm 1 and $b_{h,t}$ in Algorithm 2), which we call the *doubly decaying* bonus. We discuss this new bonus design in the following.

---

**Algorithm 2** RSQ2

---

1: $Q_h(\cdot,\cdot), V_h(\cdot) \leftarrow H - h + 1$ if $\beta > 0$; $Q_h(\cdot,\cdot), V_h(\cdot) \leftarrow 0$ otherwise, for all $h \in [H+1]$
2: $N_h(\cdot,\cdot) \leftarrow 0$ for all $h \in [H]$; $\alpha_u \leftarrow \frac{H+1}{H+u}$ for $u \in \mathbb{Z}$
3: **for** episode $k = 1, \ldots, K$ **do**
4:     Receive the initial state $s_1$
5:     **for** step $h = 1, \ldots, H$ **do**
6:         Take action $a_h \leftarrow \operatorname{argmax}_{a' \in \mathcal{A}} \frac{1}{\beta} \log\{G_h(s_h, a')\}$ and observe $r_h(s_h, a_h)$ and $s_{h+1}$
7:         Add 1 to $N_h(s_h, a_h)$;   $t \leftarrow N_h(s_h, a_h)$
8:         $b_{h,t} \leftarrow c|e^{\beta(H-h+1)} - 1|\sqrt{\frac{H \log(HSAK/\delta)}{t}}$ for some universal constant $c > 0$
9:         $w_h(s_h, a_h) \leftarrow (1 - \alpha_t) \cdot G_h(s_h, a_h) + \alpha_t \cdot e^{\beta[r_h(s_h, a_h) + V_{h+1}(s_{h+1})]}$
10:        $G_h(s_h, a_h) \leftarrow \begin{cases} \min\{w_h(s_h, a_h) + \alpha_t b_{h,t}, \ e^{\beta(H-h+1)}\}, & \text{if } \beta > 0 \\ \max\{w_h(s_h, a_h) - \alpha_t b_{h,t}, \ e^{\beta(H-h+1)}\}, & \text{if } \beta < 0 \end{cases}$
11:         $V_h(s_h) \leftarrow \max_{a' \in \mathcal{A}} \frac{1}{\beta} \log\{G_h(s_h, a')\}$
12:     **end for**
13: **end for**

---

## 5.2 Doubly decaying bonus

Let us focus on $\beta > 0$ for this discussion. In optimism-based algorithms, the bonus term is used to enforce the upper confidence bound in order to encourage sufficient exploration in uncertain environments. It takes the form of a multiplier times a factor that is inversely proportional to visit counts $\{N_h\}$. Our bonus follows this structure and is given by

$$b_h(s, a) \propto (e^{\beta(H-h+1)} - 1)\sqrt{\frac{1}{N_h(s, a)}}, \tag{16}$$

ignoring factors that do not vary in $(h, s, a)$. In Equation (16), the quantity $e^{\beta(H-h+1)}$ plays the role of the multiplier and $\sqrt{1/N_h(s, a)}$ is the factor that decreases in the visit count. While the component $\sqrt{1/N_h(s, a)}$ is common in bonus terms, our new bonus is designed to shrink its multiplier deterministically and exponentially across the horizon steps, as $e^{\beta(H-h+1)} - 1$ decreases from $e^{\beta H} - 1$ in step $h = 1$ to $e^{\beta} - 1$ in step $h = H$. This is in sharp contrast with the bonus terms typically found in risk-neutral RL algorithms, where the multipliers are kept constant in $h$ (usually as a constant multiple of $H$). Furthermore, our bonus design is also in contrast with that in RSVI and RSQ proposed by [20], whose multiplier is $e^{\beta H} - 1$ and kept fixed along the horizon. Because $b_h$ decays both in the visit count $N_h(s, a)$ (across episodes) and the multiplier $e^{\beta(H-h+1)} - 1$ (across the horizon), we name it as *doubly decaying* bonus. We remark that this is a novel feature of Algorithms 1 and 2, compared to RSVI and RSQ. Let us discuss how this new exploration mechanism is motivated from the error dynamics (14).

**Motivation of exponential decay.** From Equation (14), we see that the error of the iterate is bounded by the sum of weighted bonus terms, where the weights are of the form $e^{\beta \cdot D_h}$ and $D_h \in [0, h-1]$. Choosing $b_h \propto e^{\beta(H-h+1)} - 1$ ensures that the weighted bonus is on the order of $e^{\beta H} - 1$ at maximum. On the other hand, if we use the bonus as in [20], which is proportional to $e^{\beta H} - 1$, then we would end up with a multiplicative factor $e^{2\beta H} - 1$ in regret, which is exponentially larger than $e^{\beta H} - 1$. An alternative way to understanding the exponential decay of our bonus is as follows. At step $h$, the estimated value function is $V_h \in [0, H - h + 1]$, which implies $e^{\beta \cdot V_h} \in [1, e^{\beta(H-h+1)}]$. The iterate $G_h$ (of Algorithm 1 or 2) is used to estimate $e^{\beta \cdot Q_h^\pi}$, with its estimation error given by

$$|e^{\beta \cdot Q_h^\pi} - G_h| \approx |e^{\beta \cdot Q_h^\pi} - \widehat{\mathbb{P}}_h e^{\beta(r_h + V_{h+1})}| \leq e^{\beta(H-h+1)} - 1,$$

where $\widehat{\mathbb{P}}_h$ denotes an empirical average operator over historical data in step $h$. Therefore, the estimation error of $G_h$ shrinks exponentially across the horizon. Since bonus is used to compensate for and dominate the estimation error, the minimal order of $b_h$ required is thus $e^{\beta(H-h+1)} - 1$, which is exactly the multiplier in Equation (16).

As a passing note, we remark that the decaying multiplier is not necessary in risk-neutral RL algorithms, since the estimation error therein satisfies $|Q_h - \widehat{\mathbb{P}}_h(r_h + V_{h+1})| \leq H - h + 1$, which is upper bounded by $H$ for all $h \in [H]$. This implies that it suffices to simply set the bonus multiplier as a constant multiple of $H$. In contrast, as we have explained, the estimation error of our algorithms decays exponentially in step $h$, and an adaptive and exponentially decaying bonus is needed.

**Comparison with Bernstein-type bonus.** We also compare our bonus in Equation (16) with the Bernstein-type bonus commonly used to improve sample efficiency of risk-neutral RL algorithms [1, 27]. The Bernstein-type bonus takes the form of

$$\bar{b}_h(s, a) \propto \sqrt{\frac{H + \widehat{\mathrm{Var}}(V_{h+1})}{N_h(s, a)}} + o\left(\sqrt{\frac{1}{N_h(s, a)}}\right), \tag{17}$$

where $\widehat{\mathrm{Var}}(\cdot)$ denotes an empirical variance operator over historical data and $o(\cdot)$ denotes a vanishing term as $N_h(s, a) \to \infty$. Our bonus in Equation (16) is different from the Bernstein-type bonus in Equation (17) in mechanism: our bonus features the multiplier $e^{\beta(H-h+1)} - 1$ which decays exponentially and deterministically over $h \in [H]$, whereas the Bernstein-type bonus uses $\sqrt{H + \widehat{\mathrm{Var}}(V_{h+1})}$ as the multiplier (ignoring the vanishing term). The term $\widehat{\mathrm{Var}}(V_{h+1})$ depends on the trajectory of the learning process. Therefore the multiplier is stochastic and stays on the polynomial order of $H$ across the horizon. Moreover, it is unclear how the multiplier behaves in terms of step $h$.

# 6 Main results

In this section, we present and discuss our main theoretical results for Algorithms 1 and 2.

**Theorem 1.** *For any $\delta \in (0, 1]$, with probability at least $1 - \delta$ there exists a universal constant $c > 0$ (used in Algorithm 1), such that the regret of Algorithm 1 is bounded by*

$$\mathrm{Regret}(K) \lesssim \frac{e^{|\beta|H} - 1}{|\beta|H} \sqrt{H^4 S^2 A K \log^2(HSAK/\delta)}.$$

**Theorem 2.** *For any $\delta \in (0, 1]$, with probability at least $1 - \delta$ and when $K$ is sufficiently large, there exists a universal constant $c > 0$ (used in Algorithm 2) such that the regret of Algorithm 2 obeys*

$$\mathrm{Regret}(K) \lesssim \frac{e^{|\beta|H} - 1}{|\beta|H} \sqrt{H^3 S A K \log(HSAK/\delta)}.$$

The proof of the two theorems are provided in Appendices B and C, respectively. Note that the above results generalize those in the literature of risk-neutral RL: when $\beta \to 0$, we recover the same regret bounds of LSVI in [28] and Q-learning in [27].

Let us discuss the connections between our results and those in [20]. The work of [20] proposes two algorithms, RSVI and RSQ, that attain the regret bound

$$\mathrm{Regret}(K) \lesssim e^{|\beta|H^2} \cdot \frac{e^{|\beta|H} - 1}{|\beta|H} \sqrt{\mathrm{poly}(H) \cdot K}, \tag{18}$$

and a lower bound incurred by any algorithm

$$\mathrm{Regret}(K) \gtrsim \frac{e^{|\beta|H'} - 1}{|\beta|H} \sqrt{\mathrm{poly}(H) \cdot K}, \tag{19}$$

where $H'$ is a linear function in $H$; for simplicity of presentation, we exclude polynomial dependencies on other parameters and logarithmic factors from the two bounds. In particular, the proof of the lower bound is based on reducing an hard instance of MDP to a multi-armed bandit. It is a priori unclear whether the extra exponential factor $e^{|\beta|H^2}$ in the upper bound (18) is fundamental in the MDP setting, or is due to suboptimal analysis or algorithmic design. We would like to mention that although one trivial way of avoiding the $e^{|\beta|H^2}$ factor in the upper bound (18) is to use a sufficiently small $|\beta|$ in the algorithms of [20] (e.g., $|\beta| \leq \frac{1}{H^2}$ so that $e^{|\beta|H^2} \lesssim 1$), such a small $|\beta|$ defeats the

very purpose of have an appropriate degree of risk-sensitivity in the algorithms. Hence, an answer for *all* $\beta \neq 0$ would be desirable.

In view of Theorems 1 and 2, we see that our Algorithms 1 and 2 achieve regret bounds that are exponentially sharper than those of RSVI and RSQ. In particular, our results eliminate the $e^{|\beta|H^2}$ factor from Equation (18) thanks to the novel analysis and doubly decaying bonus in our algorithms, which are inspired by the exponential Bellman equation (5). As a result, our bounds significantly narrow the gap between upper bounds and the lower bound (19).

## Acknowledgments and Disclosure of Funding

We thank the reviewers for their constructive feedback. Z. Yang acknowledges Simons Institute (Theory of Reinforcement Learning). Y. Chen is partially supported by NSF grant CCF-1704828 and CAREER Award CCF-2047910. Z. Wang acknowledges National Science Foundation (Awards 2048075, 2008827, 2015568, 1934931), Simons Institute (Theory of Reinforcement Learning), Amazon, J.P. Morgan, and Two Sigma for their supports.

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
