Distributional RL has been studied in the line of works [3, 15, 19, 32, 34, 35, 39, 45]. The framework of distributional RL is built upon the following key equation, namely the distributional Bellman equation:

$$\forall h \in [H], \quad Z_h^\pi(s, a) \stackrel{d}{=} R_h(s, a) + Z_{h+1}^\pi(X', U'), \tag{20}$$

for a fixed policy $\pi$, where $Z_{H+1}^\pi(\cdot, \cdot) := 0$, $X' \sim P_h(\cdot \mid s, a)$, $U' \sim \pi(\cdot \mid X')$ and $R_h$ is the reward distribution in step $h$. Here, we use $\stackrel{d}{=}$ to denote equality in distribution. It can be seen that $Z_h^\pi(s, a)$ is the distribution of cumulative rewards under policy $\pi$ at step $h$, when the state and action $(s, a)$ are visited in step $h$. Based on Equation (20), a distributional Bellman optimality operator $\mathbb{T}_h$ is given by

$$[\mathbb{T}_h Z](s, a) \stackrel{d}{:=} R_h(s, a) + Z_{h+1}(X', \operatorname*{argmax}_{a' \in \mathcal{A}} \mathbb{E}[Z_{h+1}(X', a')]), \tag{21}$$

where again $X' \sim P_h(\cdot \mid s, a)$. Note that in Equation (21), the optimal action is greedy with respect to the *expectation* of the distribution $Z_{h+1}$. Most existing distributional RL algorithms work with distribution estimates such as quantiles [14, 15] or empirical distribution functions [3, 39].

Now recall the exponential Bellman equation (5), which takes the form

$$\forall h \in [H], \quad e^{\mu \cdot Q_h^\pi(s,a)} = \mathbb{E}_{X'}[e^{\mu(r_h(s,a) + V_{h+1}^\pi(X'))}], \tag{22}$$

for any fixed $\mu \in \mathbb{R}$, where $V_{H+1}^\pi(\cdot) := 0$, $X' \sim P_h(\cdot \mid s, a)$ and $r_h$ is the deterministic reward function by our assumption. Given the definitions (1) and (2) with $\beta$ replaced by $\mu$, we note that both $Q_h^\pi$ and $V_{h+1}^\pi$ in the above equation depend on the value of $\mu$ (which we omit for simplicity of notations). Then by the definition of $Q_h^\pi$ in Equation (2), one sees that $\{e^{\mu \cdot Q_h^\pi} : \mu \in \mathbb{R}\}$ represents the MGF of the cumulative rewards at step $h$ when policy $\pi$ is executed. Hence, the exponential Bellman equation for risk-sensitive RL provides an instantiation of Equation (20) through the MGF of rewards.

## B  Proof of Theorem 1

First, we set some notations and definitions. Define $\iota := \log(2HSAK/\delta)$ for a given $\delta \in (0, 1]$. We adopt the shorthands $\mathbb{I}_h^\tau(s, a) := \mathbb{I}\{(s_h^\tau, a_h^\tau) = (s, a)\}$ and $r_h^\tau := r_h(s_h^\tau, a_h^\tau)$ for $(\tau, h) \in [K] \times [H]$. We let $N_h^k(s, a)$ be the visit count of $(h, s, a)$ at the beginning of episode $k$. We denote by $V_h^k$, $G_h^k$, $b_h^k$ the values of $V_h$, $G_h$, $b_h$ after the updates in step $h$ of episode $k$, respectively. We also set $Q_h^k = \frac{1}{\beta} \log\{G_h^k\}$.

For the time being we consider $\beta > 0$. For $h \in [H]$, we define

$$\delta_h^k := e^{\beta V_h^k(s_h^k)} - e^{\beta V_h^{\pi^k}(s_h^k)},$$

$$\zeta_{h+1}^k := [P_h(e^{\beta[r_h(s_h^k, a_h^k) + V_{h+1}^k(s')]} - e^{\beta[r_h(s_h^k, a_h^k) + V_{h+1}^{\pi^k}(s')]})](s_h^k, a_h^k) - e^{\beta r_h(s_h^k, a_h^k)} \delta_{h+1}^k,$$

where $[P_h f](s, a) := \mathbb{E}_{s' \sim P_h(\cdot | s, a)}[f(s')]$ for any $f : \mathcal{S} \to \mathbb{R}$ and $(s, a) \in \mathcal{S} \times \mathcal{A}$. It can be seen that $b_h^k$ in Algorithm 1 can be equivalently defined as

$$b_h^k := c(e^{\beta(H-h+1)} - 1)\sqrt{\frac{S\iota}{\max\{1, N_h^k(s_h^k, a_h^k)\}}}, \tag{23}$$

where $c$ is the universal constant from Lemma 2. For any $(k, h) \in [K] \times [H]$, we have

$$\delta_h^k \stackrel{(i)}{=} (e^{\beta \cdot Q_h^k} - e^{\beta \cdot Q_h^{\pi^k}})(s_h^k, a_h^k)$$

$$\overset{(ii)}{=} \left[\min\{e^{\beta(H-h+1)}, (w_h^k + b_h^k)(s_h^k, a_h^k)\} - \mathbb{E}_{s' \sim P_h(\cdot \mid s_h^k, a_h^k)} e^{\beta[r_h(s_h^k, a_h^k) + V_{h+1}^k(s')]}\right]$$

$$+ \left[\mathbb{E}_{s' \sim P_h(\cdot \mid s_h^k, a_h^k)} e^{\beta[r_h(s_h^k, a_h^k) + V_{h+1}^k(s')]} - \mathbb{E}_{s' \sim P_h(\cdot \mid s_h^k, a_h^k)} e^{\beta[r_h(s_h^k, a_h^k) + V_{h+1}^{\pi^k}(s')]}\right]$$

$$\overset{(iii)}{\leq} 2b_h^k + [P_h(e^{\beta[r_h(s_h^k, a_h^k) + V_{h+1}^k(s')]} - e^{\beta[r_h(s_h^k, a_h^k) + V_{h+1}^{\pi^k}(s')]})](s_h^k, a_h^k)$$

$$= 2b_h^k + e^{\beta \cdot r_h(s_h^k, a_h^k)}\delta_{h+1}^k + \zeta_{h+1}^k \tag{24}$$

In the above equation, step $(i)$ holds by the construction of Algorithm 1 and the definition of $V_h^{\pi^k}$ in Equation (3); step $(ii)$ holds by Equations (29) and (30). step $(iii)$ holds on the event of Lemma 2; the last step follows from Lemma 4.

Using the fact that $V_{H+1}^k(s) = V_{H+1}^{\pi^k}(s) = 0$ and that $r_h(\cdot, \cdot) \in [0, 1]$, we can expand the recursion in Equation (24) and get

$$\delta_1^k \leq \sum_{h \in [H]} e^{\beta(h-1)}\zeta_{h+1}^k + 2\sum_{h \in [H]} e^{\beta(h-1)}b_h^k.$$

Summing the above display over $k \in [K]$ gives

$$\sum_{k \in [K]} \delta_1^k \leq \sum_{k \in [K]}\sum_{h \in [H]} e^{\beta(h-1)}\zeta_{h+1}^k + 2\sum_{k \in [K]}\sum_{h \in [H]} e^{\beta(h-1)}b_h^k \tag{25}$$

Let us now control the two terms in Equation (25). Note that $\{\zeta_{h+1}^k\}$ is a martingale difference sequence satisfying $|\zeta_h^k| \leq 2H$ for all $(k, h) \in [K] \times [H]$. By the Azuma-Hoeffding inequality, we have for any $t > 0$,

$$\mathbb{P}\left(\sum_{k \in [K]}\sum_{h \in [H]} e^{\beta(h-1)}\zeta_{h+1}^k \geq t\right) \leq \exp\left(-\frac{t^2}{2HK(e^{\beta H} - 1)^2}\right).$$

Hence, with probability $1 - \delta/2$, there holds

$$\sum_{k \in [K]}\sum_{h \in [H]} e^{\beta(h-1)}\zeta_{h+1}^k \leq (e^{\beta H} - 1)\sqrt{2HK\log(2/\delta)} \leq (e^{\beta H} - 1)\sqrt{2HK\iota}, \tag{26}$$

where $\iota = \log(2HSAK/\delta)$. For the second term in Equation (25), recall the definition of $b_h^k$ in Equation (23), and we can derive

$$\sum_{k \in [K]}\sum_{h \in [H]} e^{\beta(h-1)}b_h^k \leq \sum_{k \in [K]}\sum_{h \in [H]} c(e^{\beta H} - 1)\sqrt{S\iota}\sqrt{\frac{1}{\max\{1, N_h^k(s_h^k, a_h^k)\}}}$$

$$= c(e^{\beta H} - 1)\sqrt{S\iota}\sum_{k \in [K]}\sum_{h \in [H]}\sqrt{\frac{1}{\max\{1, N_h^k(s_h^k, a_h^k)\}}}$$

$$\overset{(i)}{\leq} c(e^{\beta H} - 1)\sqrt{S\iota}\sum_{h \in [H]}\sqrt{K}\sqrt{\sum_{k \in [K]}\frac{1}{\max\{1, N_h^k(s_h^k, a_h^k)\}}}$$

$$\leq c(e^{\beta H} - 1)\sqrt{S\iota}\sqrt{2H^2SAK\iota}, \tag{27}$$

where step $(i)$ follows the Cauchy-Schwarz inequality and the last step holds by the pigeonhole principle. Plugging Equations (26) and (27) back to Equation (25) yields

$$\sum_{k \in [K]} \delta_1^k \leq (e^{\beta H} - 1)\sqrt{2HK\iota} + 2c(e^{\beta H} - 1)\sqrt{2H^2S^2AK\iota^2}$$

$$\lesssim (e^{\beta H} - 1)\sqrt{2H^2S^2AK\iota^2},$$

The proof for $\beta > 0$ is completed by invoking Lemma 9 on the event of Lemma 4. We note that the proof of $\beta < 0$ follows a similar procedure and is therefore omitted.

## B.1 Auxiliary lemmas

Let us fix a pair $(s, a) \in \mathcal{S} \times \mathcal{A}$. Recall from Algorithm 1 that

$$w_h^k(s, a) = \frac{1}{N_h^k(s, a)} \sum_{\tau \in [k-1]} \mathbb{I}_h^\tau(s, a) \left[ e^{\beta[r_h^\tau + V_{h+1}^k(s_{h+1}^\tau)]} \right]. \tag{28}$$

If $N_h^k(s, a) \geq 1$, we define

$$q_{h,1}^{k,+}(s, a) := \begin{cases} w_h^k(s, a) + b_h^k(s, a), & \text{if } \beta > 0, \\ w_h^k(s, a) - b_h^k(s, a), & \text{if } \beta < 0. \end{cases}$$

$$q_{h,1}^k(s, a) := \begin{cases} \min\{q_{h,1}^{k,+}(s, a), e^{\beta(H-h+1)}\}, & \text{if } \beta > 0, \\ \max\{q_{h,1}^{k,+}(s, a), e^{\beta(H-h+1)}\}, & \text{if } \beta < 0, \end{cases}$$

and if $N_h^k(s, a) = 0$, we let

$$q_{h,1}^{k,+}(s, a) = q_{h,1}^k(s, a) := e^{\beta(H-h+1)}.$$

Also define

$$q_{h,2}^k(s, a) := \begin{cases} \frac{1}{N_h^k(s,a)} \sum_{\tau \in [k-1]} \mathbb{I}_h^\tau(s, a) \left[ \mathbb{E}_{s' \sim P_h(\cdot \mid s_h^\tau, a_h^\tau)} e^{\beta[r_h^\tau + V_{h+1}^k(s')]} \right], & \text{if } N_h^k(s, a) \geq 1, \\ e^{\beta(H-h+1)} \text{ if } \beta > 0; \qquad 1 \text{ if } \beta < 0, & \text{if } N_h^k(s, a) = 0, \end{cases}$$

and for any policy $\pi$,

$$q_{h,3}^{k,\pi}(s, a) := \begin{cases} \frac{1}{N_h^k(s,a)} \sum_{\tau \in [k-1]} \mathbb{I}_h^\tau(s, a) \left[ \mathbb{E}_{s' \sim P_h(\cdot \mid s_h^\tau, a_h^\tau)} e^{\beta[r_h^\tau + V_{h+1}^\pi(s')]} \right], & \text{if } N_h^k(s, a) \geq 1, \\ e^{\beta \cdot Q_h^\pi(s,a)} & \text{if } N_h^k(s, a) = 0. \end{cases}$$

It can be seen that

$$q_{h,2}^k(s, a) = \mathbb{E}_{s' \sim P_h(\cdot \mid s,a)} e^{\beta[r_h(s,a) + V_{h+1}^k(s')]} \tag{29}$$

when $N_h^k(s, a) \geq 1$, and

$$q_{h,3}^{k,\pi}(s, a) = e^{\beta Q_h^\pi(s,a)} = \mathbb{E}_{s' \sim P_h(\cdot \mid s,a)} e^{\beta[r_h(s,a) + V_{h+1}^\pi(s')]} \tag{30}$$

for all $(k, h, s, a) \in [K] \times [H] \times \mathcal{S} \times \mathcal{A}$ by the exponential Bellman equation (5). We have that if $\beta > 0$,

$$e^{\beta Q_h^k} - e^{\beta Q_h^\pi} = q_{h,1}^k - q_{h,3}^{k,\pi} = (q_{h,1}^k - q_{h,2}^k) + (q_{h,2}^k - q_{h,3}^{k,\pi}), \tag{31}$$

and if $\beta < 0$,

$$e^{\beta Q_h^\pi} - e^{\beta Q_h^k} = q_{h,3}^{k,\pi} - q_{h,1}^k = (q_{h,3}^{k,\pi} - q_{h,2}^k) + (q_{h,2}^k - q_{h,1}^k). \tag{32}$$

Let us state a uniform concentration result.

**Lemma 1.** *Define $\iota := \log(2HSAK/\delta)$ and*

$$\bar{\mathcal{V}}_{h+1} := \left\{ \bar{V}_{h+1} : \mathcal{S} \to \mathbb{R} \mid \forall s \in \mathcal{S}, \ \bar{V}_{h+1}(s) \in [0, H-h] \right\}.$$

*For any $\delta \in (0, 1]$, there extsts a universal constant $c_0 > 0$ such that with probability $1 - \delta$, we have*

$$\left| \frac{1}{N_h^k(s, a)} \sum_{\tau \in [k-1]} \mathbb{I}_h^\tau(s, a) \left[ e^{\beta[r_h^\tau + \bar{V}(s_{h+1}^\tau)]} - \mathbb{E}_{s' \sim P_h(\cdot \mid s_h^\tau, a_h^\tau)} e^{\beta[r_h^\tau + \bar{V}(s')]} \right] \right|$$

$$\leq c_0 (e^{\beta(H-h+1)} - 1) \sqrt{\frac{S\iota}{\max\{1, N_h^k(s, a)\}}}$$

*for all $\bar{V} \in \bar{\mathcal{V}}_{h+1}$ and all $(k, h, s, a) \in [K] \times [H] \times \mathcal{S} \times \mathcal{A}$ that satisfies $N_h^k(s, a) \geq 1$.*

*Proof.* The result is a simple adaptation of [20, Lemma 6]. $\square$

We now control the difference $q_{h,1}^k - q_{h,2}^k$.

**Lemma 2.** *Recall the definition of $b_h^k$ from Algorithm 1. For all $(k,h,s,a) \in [K] \times [H] \times \mathcal{S} \times \mathcal{A}$, there exists some universal constant $c > 0$ (where $c$ is used in Line 6 of Algorithm 1) such that the following holds with probability at least $1 - \delta/2$: if $\beta > 0$, we have*

$$0 \leq (q_{h,1}^k - q_{h,2}^k)(s,a) \leq 2b_h^k,$$

*and if $\beta < 0$, we have*

$$0 \leq (q_{h,2}^k - q_{h,1}^k)(s,a) \leq 2b_h^k.$$

*Proof.* Let us fix a tuple $(k,h,s,a) \in [K] \times [H] \times \mathcal{S} \times \mathcal{A}$.

**Case $\beta > 0$.** For $N_h^k(s,a) = 0$, we have $q_{h,1}^k \leq e^{\beta(H-h+1)}$ and $q_{h,2}^k \geq 1$ by construction and the result follows immediately. Now we assume $N_h^k(s,a) \geq 1$. By Equation (28) we can compute

$$\left| (q_{h,1}^{k,+} - b_h^k - q_{h,2}^k)(s,a) \right|$$

$$= \left| \frac{1}{N_h^k(s,a)} \sum_{\tau \in [k-1]} \mathbb{I}_h^\tau(s,a) \left[ e^{\beta[r_h^\tau + V_{h+1}^k(s_{h+1}^\tau)]} - \mathbb{E}_{s' \sim P_h(\cdot \,|\, s_h^\tau, a_h^\tau)} \left[ e^{\beta[r_h^\tau + V_{h+1}^k(s')]} \right] \right] \right|$$

$$\leq c_0(e^{\beta(H-h+1)} - 1) \sqrt{\frac{S\iota}{\max\{1, N_h^k(s,a)\}}},$$

where the last step holds by Lemma 1 with $c_0 > 0$ being a universal constant. Setting $c$ in $b_h^k$ to be equal to $c_0$, we have

$$0 \leq (q_{h,1}^{k,+} - q_{h,2}^k)(s,a) \leq 2b_h^k.$$

Therefore, we have $q_{h,1}^k \geq q_{h,2}^k$ by the first inequality above, the definition of $q_{h,1}^k$ and the property $q_{h,2}^k \leq e^{\beta(H-h+1)}$. Also, since $q_{h,1}^{k,+} \geq q_{h,1}^k$, it holds that $q_{h,1}^k - q_{h,2}^k \leq q_{h,1}^{k,+} - q_{h,2}^k$. The conclusion follows.

**Case $\beta < 0$.** We have, similar to the previous case, that

$$\left| (q_{h,1}^{k,+} - b_h^k - q_{h,2}^k)(s,a) \right| \leq c_0(1 - e^{\beta(H-h+1)}) \sqrt{\frac{S\iota}{\max\{1, N_h^k(s,a)\}}}.$$

Choosing $c = c_0$ in the definition of $b_h^k(s,a)$ leads to

$$0 \leq (q_{h,2}^k - q_{h,1}^{k,+})(s,a) \leq 2b_h^k.$$

This implies $q_{h,2}^k \geq q_{h,1}^{k,+}$, and since $q_{h,1}^k, q_{h,2}^k \geq e^{\beta(H-h+1)}$, we also have $q_{h,2}^k \geq q_{h,1}^k$. In addition, since $q_{h,1}^{k,+} \leq q_{h,1}^k$, it also holds that $q_{h,2}^k - q_{h,1}^k \leq q_{h,2}^k - q_{h,1}^{k,+}$. Then the conclusion of this case follows. $\square$

**Lemma 3.** *On the event of Lemma 2, for all $(k,h,s,a) \in [K] \times [H] \times \mathcal{S} \times \mathcal{A}$ and any policy $\pi$, we have*

$$\begin{cases} e^{\beta \cdot Q_h^k(s,a)} \geq e^{\beta \cdot Q_h^\pi(s,a)}, & \text{if } \beta > 0, \\ e^{\beta \cdot Q_h^k(s,a)} \leq e^{\beta \cdot Q_h^\pi(s,a)}, & \text{if } \beta < 0. \end{cases}$$

*Proof.* We focus on the case of $\beta > 0$ since the proof for $\beta < 0$ is very similar. For the purpose of the proof, we set $Q_{H+1}^\pi(s,a) = Q_{H+1}^*(s,a) = 0$ for all $(s,a) \in \mathcal{S} \times \mathcal{A}$. We fix a tuple $(k,s,a) \in [K] \times \mathcal{S} \times \mathcal{A}$ and use strong induction on $h$. The base case for $h = H+1$ is satisfied since $e^{\beta \cdot Q_{H+1}^k(s,a)} = e^{\beta \cdot Q_{H+1}^\pi(s,a)} = 1$ for $k \in [K]$ by definition. Now we fix an $h \in [H]$ and assume that $e^{\beta \cdot Q_{h+1}^k(s,a)} \geq e^{\beta \cdot Q_{h+1}^*(s,a)}$. Moreover, by the induction assumption we have

$$e^{\beta \cdot V_{h+1}^k(s)} = \max_{a' \in \mathcal{A}} e^{\beta \cdot Q_{h+1}^k(s,a')} \geq \max_{a' \in \mathcal{A}} e^{\beta \cdot Q_{h+1}^\pi(s,a')} \geq e^{\beta \cdot V_{h+1}^\pi(s)}. \tag{33}$$

We also assume that $(s,a)$ satisfies $N_h^k(s,a) \geq 1$, since otherwise $e^{\beta \cdot Q_h^k(s,a)} = e^{\beta(H-h+1)} \geq e^{\beta \cdot Q_h^\pi(s,a)}$ and we are done. This assumption and Equation (33) together imply $q_{h,2}^k \geq q_{h,3}^{k,\pi}$ by Lemma 2. We also have $q_{h,1}^k \geq q_{h,2}^k$ on the event of Lemma 2. Therefore, it follows that $e^{\beta \cdot Q_h^k(s,a)} \geq e^{\beta \cdot Q_h^\pi(s,a)}$ by Equation (31) and we have completed the induction. $\square$

**Lemma 4.** *For all $(k, h, s) \in [K] \times [H] \times \mathcal{S}$ and any $\delta \in (0, 1]$, with probability at least $1 - \delta/2$, we have*

$$\begin{cases} e^{\beta \cdot V_h^k(s)} \geq e^{\beta \cdot V_h^\pi(s)}, & \text{if } \beta > 0, \\ e^{\beta \cdot V_h^k(s)} \leq e^{\beta \cdot V_h^\pi(s)}, & \text{if } \beta < 0. \end{cases}$$

*Proof.* The result follows from Lemma 3 and Equation (33). $\qquad\square$

## C   Proof of Theorem 2

We first lay out some additional notations to facilitate our proof. Let $N_h^k$, $G_h^k$, $V_h^k$ be the $N_h$, $G_h$, $V_h$ functions at the beginning of the episode $k$, before $t$ is updated. We also set $Q_h^k := \frac{1}{\beta} G_h^k$. We let $\widehat{P}_h^k(\cdot \mid s, a)$ denote the delta function centered at $s_{h+1}^k$ for all $(k, h, s, a) \in [K] \times [H] \times \mathcal{S} \times \mathcal{A}$. This means $\mathbb{E}_{s' \sim \widehat{P}_h^k(\cdot \mid s, a)}[f(s')] = f(s_{h+1}^k)$ for any $f : \mathcal{S} \to \mathbb{R}$. Denote by $n_h^k := N_h^k(s_h^k, a_h^k)$. Recall from Algorithm 2, the learning rate is defined as

$$\alpha_t := \frac{H + 1}{H + t}, \tag{34}$$

for $t \in \mathbb{Z}$.

For now we consider the case for $\beta > 0$. We define the following quantities to ease the notations for the proof:

$$\delta_h^k := e^{\beta \cdot V_h^k(s_h^k)} - e^{\beta \cdot V_h^{\pi^k}(s_h^k)},$$

$$\phi_h^k := e^{\beta \cdot V_h^k(s_h^k)} - e^{\beta \cdot V_h^*(s_h^k)},$$

$$\xi_{h+1}^k := [(P_h - \widehat{P}_h^k)(e^{\beta \cdot V_{h+1}^*} - e^{\beta \cdot V_{h+1}^{\pi^k}})](s_h^k, a_h^k).$$

For each fixed $(k, h) \in [K] \times [H]$, we let $t = N_h^k(s_h^k, a_h^k)$. Then it holds that

$$\delta_h^k \overset{(i)}{=} e^{\beta \cdot Q_h^k(s_h^k, a_h^k)} - e^{\beta \cdot Q_h^{\pi^k}(s_h^k, a_h^k)}$$

$$= [e^{\beta \cdot Q_h^k(s_h^k, a_h^k)} - e^{\beta \cdot Q_h^*(s_h^k, a_h^k)}] + [e^{\beta \cdot Q_h^*(s_h^k, a_h^k)} - e^{\beta \cdot Q_h^{\pi^k}(s_h^k, a_h^k)}]$$

$$\overset{(ii)}{=} [e^{\beta \cdot Q_h^k(s_h^k, a_h^k)} - e^{\beta \cdot Q_h^*(s_h^k, a_h^k)}] + e^{\beta \cdot r_h(s_h^k, a_h^k)}[P_h(e^{\beta \cdot V_{h+1}^*} - e^{\beta \cdot V_{h+1}^{\pi^k}})](s_h^k, a_h^k)$$

$$\overset{(iii)}{\leq} [e^{\beta \cdot Q_h^k(s_h^k, a_h^k)} - e^{\beta \cdot Q_h^*(s_h^k, a_h^k)}] + e^{\beta}[P_h(e^{\beta \cdot V_{h+1}^*} - e^{\beta \cdot V_{h+1}^{\pi^k}})](s_h^k, a_h^k)$$

$$= [e^{\beta \cdot Q_h^k(s_h^k, a_h^k)} - e^{\beta \cdot Q_h^*(s_h^k, a_h^k)}] + e^{\beta}(\delta_{h+1}^k - \phi_{h+1}^k + \xi_{h+1}^k)$$

$$\overset{(iv)}{\leq} \alpha_t^0(e^{\beta(H-h+1)} - 1) + 2\gamma_{h,t} + \sum_{i \in [t]} \alpha_t^i \cdot e^{\beta}\left[e^{\beta \cdot V_{h+1}^{k_i}(s_{h+1}^{k_i})} - e^{\beta \cdot V_{h+1}^*(s_{h+1}^{k_i})}\right]$$

$$\quad + e^{\beta}(\delta_{h+1}^k - \phi_{h+1}^k + \xi_{h+1}^k)$$

$$= \alpha_t^0(e^{\beta(H-h+1)} - 1) + 2\gamma_{h,t} + \sum_{i \in [t]} \alpha_t^i \cdot e^{\beta} \phi_{h+1}^{k_i}$$

$$\quad + e^{\beta}(\delta_{h+1}^k - \phi_{h+1}^k + \xi_{h+1}^k) \tag{35}$$

where step $(i)$ holds since $V_h^k(s_h^k) = \max_{a' \in \mathcal{A}} Q_h^k(s_h^k, a') = Q_h^k(s_h^k, a_h^k)$ and $V_h^{\pi^k}(s_h^k) = Q_h^{\pi^k}(s_h^k, \pi_h^k(s_h^k)) = Q_h^{\pi^k}(s_h^k, a_h^k)$; step $(ii)$ holds by the exponential Bellman equation (5); step $(iii)$ holds since $V_{h+1}^* \geq V_{h+1}^{\pi^k}$ implies $e^{\beta \cdot V_{h+1}^*} \geq e^{\beta \cdot V_{h+1}^{\pi^k}}$ given that $\beta > 0$; step $(iv)$ holds on the event of Lemma 8 (with $\gamma_{h,t}$ defined therein).

We bound each term in (35) one by one. First, we have

$$\sum_{k \in [K]} \alpha_{n_h^k}^0(e^{\beta(H-h+1)} - 1) = (e^{\beta(H-h+1)} - 1) \sum_{k \in [K]} \mathbb{I}\{n_h^k = 0\}$$

$$\leq (e^{\beta(H-h+1)} - 1)SA.$$

The second term in (35) can be bounded by

$$\sum_{k\in[K]} \left( \sum_{i\in[t]} \alpha_t^i \cdot e^\beta \phi_{h+1}^{k_i} \right) = \sum_{k\in[K]} \left( \sum_{i\in[n_h^k]} \alpha_{n_h^k}^i \cdot e^\beta \phi_{h+1}^{k_i(s_h^k,a_h^k)} \right),$$

where $k_i(s_h^k, a_h^k)$ denotes the episode in which $(s_h^k, a_h^k)$ was taken at step $h$ for the $i$-th time. We re-group the above summation in a different way. For every $k' \in [K]$, the term $\phi_{h+1}^{k'}$ appears in the summand with $k > k'$ if and only if $(s_h^k, a_h^k) = (s_h^{k'}, a_h^{k'})$. For the first time we visit $(s_h^{k'}, a_h^{k'})$ we have $n_h^k = n_h^{k'} + 1$, for the second time we have $n_h^k = n_h^{k'} + 2$, and etc. Therefore, we may continue the above display as

$$\sum_{k\in[K]} \left( \sum_{i\in[n_h^k]} \alpha_{n_h^k}^i \cdot e^\beta \phi_{h+1}^{k_i(s_h^k,a_h^k)} \right) \leq \sum_{k'\in[K]} e^\beta \phi_{h+1}^{k'} \left( \sum_{t\geq n_h^{k'}+1} \alpha_t^{n_h^{k'}} \right)$$

$$\leq \left(1 + \frac{1}{H}\right) e^\beta \sum_{k'\in[K]} \phi_{h+1}^{k'},$$

where the last step follows Fact 1(c). Collecting the above results and plugging them into Equation (35), we have

$$\sum_{k\in[K]} \delta_h^k \leq (e^{\beta(H-h+1)} - 1)SA + \left(1 + \frac{1}{H}\right) e^\beta \sum_{k\in[K]} \phi_{h+1}^k$$

$$+ \sum_{k\in[K]} e^\beta(\delta_{h+1}^k - \phi_{h+1}^k + \xi_{h+1}^k) + 2\sum_{k\in[K]} \gamma_{h,n_h^k}$$

$$\leq (e^{\beta(H-h+1)} - 1)SA + \left(1 + \frac{1}{H}\right) e^\beta \sum_{k\in[K]} \delta_{h+1}^k$$

$$+ \sum_{k\in[K]} (2\gamma_{h,n_h^k} + e^\beta \xi_{h+1}^k), \tag{36}$$

where the last step holds since $\delta_{h+1}^k \geq \phi_{h+1}^k$ (due to the fact that $\beta > 0$ and $V_{h+1}^* \geq V_{h+1}^{\pi^k}$).

Now, we unroll the quantity $\sum_{k\in[K]} \delta_h^k$ recursively in the form of Equation (36), and get

$$\sum_{k\in[K]} \delta_1^k \leq \sum_{h\in[H]} \left[ \left(1 + \frac{1}{H}\right) e^\beta \right]^{h-1} \left[ (e^{\beta(H-h+1)} - 1)SA + \sum_{k\in[K]} (2\gamma_{h,n_h^k} + e^\beta \xi_{h+1}^k) \right]$$

$$= \sum_{h\in[H]} \left(1 + \frac{1}{H}\right)^{h-1} \left[ (e^{\beta H} - e^{\beta(h-1)})SA + \sum_{k\in[K]} (2e^{\beta(h-1)}\gamma_{h,n_h^k} + e^{\beta h}\xi_{h+1}^k) \right]$$

$$= \sum_{h\in[H]} \left(1 + \frac{1}{H}\right)^{h-1} \left[ (e^{\beta H} - e^{\beta(h-1)})SA + \sum_{k\in[K]} 2e^{\beta(h-1)}\gamma_{h,n_h^k} \right]$$

$$+ \sum_{h\in[H]} \sum_{k\in[K]} \left(1 + \frac{1}{H}\right)^{h-1} e^{\beta h}\xi_{h+1}^k$$

$$\leq e \left[ (e^{\beta H} - 1)HSA + \sum_{k\in[K]} \sum_{h\in[H]} 2e^{\beta(h-1)}\gamma_{h,n_h^k} \right] + \sum_{h\in[H]} \sum_{k\in[K]} \left(1 + \frac{1}{H}\right)^{h-1} e^{\beta h}\xi_{h+1}^k, \tag{37}$$

where the first step uses the fact that $\delta_{H+1}^k = 0$ for $k \in [K]$; the last step holds since $(1 + 1/H)^h \leq (1 + 1/H)^H \leq e$ for all $h \in [H]$. By the pigeonhole principle, for any $h \in [H]$ we have

$$\sum_{k \in [K]} \sum_{h \in [H]} e^{\beta(h-1)} \gamma_{n_h^k} \lesssim (e^{\beta H} - 1) \sum_{k \in [K]} \sqrt{\frac{H\iota}{n_h^k}}$$

$$\lesssim (e^{\beta H} - 1) \sum_{(s,a) \in \mathcal{S} \times \mathcal{A}} \sum_{n \in [N_h^K(s,a)]} \sqrt{\frac{H\iota}{n}}$$

$$\lesssim (e^{\beta H} - 1)\sqrt{HSAK\iota} \tag{38}$$

where the third step holds since $\sum_{(s,a) \in \mathcal{S} \times \mathcal{A}} N_h^K(s,a) = K$ and the RHS of the second step is maximized when $N_h^K(s,a) = K/(SA)$ for all $(s,a) \in \mathcal{S} \times \mathcal{A}$. Finally, the Azuma-Hoeffding inequality and the fact that $\left|\left(1 + \frac{1}{H}\right)^{h-1} e^{\beta h}\xi_{h+1}^k\right| \leq e(e^{\beta H} - 1)$ for $h \in [H]$ together imply that with probability at least $1 - \delta$, we have

$$\left|\sum_{h \in [H]} \sum_{k \in [K]} \left(1 + \frac{1}{H}\right)^{h-1} e^{\beta h}\xi_{h+1}^k\right| \lesssim (e^{\beta H} - 1)\sqrt{HK\iota}. \tag{39}$$

Plugging Equations (38) and (39) into (37), we have

$$\sum_{k \in [K]} \delta_1^k \lesssim (e^{\beta H} - 1)\sqrt{HSAK\iota},$$

when $K$ is large enough. Invoking Lemma 9 completes the proof for the case $\beta > 0$.

The proof is very similar for the case of $\beta < 0$, and one only needs to exchange the role of $V_h^k$ and $V_h^{\pi^k}$ in the definitions of $\delta_h^k$, $\phi_h^k$, $\xi_h^k$, etc, to get the counterpart of Equation (35) and of the remaining analysis.

## C.1 Auxiliary lemmas

Recall the learning rate $\alpha_t$ defined in Equation (34). We define

$$\alpha_t^0 := \prod_{j=1}^t (1 - \alpha_j), \qquad \alpha_t^i := \alpha_i \prod_{j=i+1}^t (1 - \alpha_j) \tag{40}$$

for integers $i, t \geq 1$. We set $\alpha_t^0 = 1$ and $\sum_{i \in [t]} \alpha_t^i = 0$ if $t = 0$, and $\alpha_t^i = \alpha_i$ if $t < i + 1$.

In the following, we provide some useful facts about the learning rate.

**Fact 1.** *The following properties hold for $\alpha_t^i$.*

(a) $\frac{1}{\sqrt{t}} \leq \sum_{i \in [t]} \frac{\alpha_t^i}{\sqrt{i}} \leq \frac{2}{\sqrt{t}}$ *for every integer $t \geq 1$.*

(b) $\max_{i \in [t]} \alpha_t^i \leq \frac{2H}{t}$ *and* $\sum_{i \in [t]} (\alpha_t^i)^2 \leq \frac{2H}{t}$ *for every integer $t \geq 1$.*

(c) $\sum_{t=i}^{\infty} \alpha_t^i = 1 + \frac{1}{H}$ *for every integer $i \geq 1$.*

(d) $\sum_{i \in [t]} \alpha_t^i = 1$ *and* $\alpha_t^0 = 0$ *for every integer $t \geq 1$, and* $\sum_{i \in [t]} \alpha_t^i = 0$ *and* $\alpha_t^0 = 1$ *for* $t = 0$.

*Proof.* The first three facts can be found in [27, Lemma 4.1], and the last one follows from direct calculation in view of Equation (40). $\square$

Define the shorthand $\iota := \log(SAT/\delta)$ for $\delta \in (0, 1]$. We fix a tuple $(k, h, s, a) \in [K] \times [H] \times \mathcal{S} \times \mathcal{A}$ with $k_i \leq k$ being the episode in which $(s, a)$ is visited the $i$-th time at step $h$. Let us define

$$q_{h,1}^{k,+}(s,a) := \alpha_t^0 e^{\beta(H-h+1)} + \begin{cases} \sum_{i \in [t]} \alpha_t^i \left[e^{\beta[r_h(s,a)+V_{h+1}^{k_i}(s_{h+1}^{k_i})]} + b_{h,i}\right], & \text{if } \beta > 0, \\ \sum_{i \in [t]} \alpha_t^i \left[e^{\beta[r_h(s,a)+V_{h+1}^{k_i}(s_{h+1}^{k_i})]} - b_{h,i}\right], & \text{if } \beta < 0, \end{cases}$$

$$q_{h,1}^k(s,a) := \begin{cases} \min\{q_{h,1}^{k,+}(s,a), e^{\beta(H-h+1)}\}, & \text{if } \beta > 0, \\ \max\{q_{h,1}^{k,+}(s,a), e^{\beta(H-h+1)}\}, & \text{if } \beta < 0, \end{cases}$$

and

$$q_{h,2}^{k,\circ}(s,a) := \alpha_t^0 e^{\beta(H-h+1)} + \sum_{i \in [t]} \alpha_t^i \left[ e^{\beta[r_h(s,a)+V_{h+1}^*(s_{h+1}^{k_i})]} \right]$$

$$q_{h,2}^{k,+}(s,a) := \alpha_t^0 e^{\beta(H-h+1)} + \begin{cases} \sum_{i \in [t]} \alpha_t^i \left[ e^{\beta[r_h(s,a)+V_{h+1}^*(s_{h+1}^{k_i})]} + b_{h,i} \right], & \text{if } \beta > 0, \\ \sum_{i \in [t]} \alpha_t^i \left[ e^{\beta[r_h(s,a)+V_{h+1}^*(s_{h+1}^{k_i})]} - b_{h,i} \right], & \text{if } \beta < 0, \end{cases}$$

$$q_{h,2}^k(s,a) := \begin{cases} \min\{q_{h,2}^{k,+}(s,a), e^{\beta(H-h+1)}\}, & \text{if } \beta > 0, \\ \max\{q_{h,2}^{k,+}(s,a), e^{\beta(H-h+1)}\}, & \text{if } \beta < 0, \end{cases}$$

and

$$q_{h,3}^k(s,a) := \alpha_t^0 e^{\beta \cdot Q_h^*(s,a)} + \sum_{i \in [t]} \alpha_t^i \left[ \mathbb{E}_{s' \sim P_h(\cdot \mid s,a)} e^{\beta[r_h(s,a)+V_{h+1}^*(s')]} \right].$$

We have a simple fact on $q_{h,2}^k$ and $q_{h,2}^{k,\circ}$.

**Fact 2.** *If $\beta > 0$, we have $q_{h,2}^{k,\circ}(\cdot,\cdot) \leq q_{h,2}^k(\cdot,\cdot)$; if $\beta < 0$, we have $q_{h,2}^{k,\circ}(\cdot,\cdot) \geq q_{h,2}^k(\cdot,\cdot)$.*

*Proof.* We focus on the case where $\beta > 0$ and the case for $\beta < 0$ can be proved similarly. Note that $r_h(s,a) + V_{h+1}^*(s_{h+1}^{k_i}) \in [0, H-h+1]$ implies $e^{\beta[r_h(s,a)+V_{h+1}^*(s_{h+1}^{k_i})]} \leq e^{\beta(H-h+1)}$. We also have $\alpha_t^0, \sum_{i \in [t]} \alpha_t^i \in \{0,1\}$ with $\alpha_t^0 + \sum_{i \in [t]} \alpha_t^i = 1$ by Fact 1(d). Together they imply that $q_{h,2}^{k,\circ}(\cdot,\cdot) \leq e^{\beta(H-h+1)}$ and $(q_{h,2}^{k,\circ} - q_{h,2}^{k,+})(\cdot,\cdot) = -\sum_{i \in [t]} \alpha_t^i b_{h,i} \leq 0$ by definition of $b_{h,i}$ in Line 8 of Algorithm 2. Therefore, $q_{h,2}^{k,\circ}(\cdot,\cdot) \leq \min\{e^{\beta(H-h+1)}, q_{h,2}^{k,+}(\cdot,\cdot)\} = q_{h,2}^k(\cdot,\cdot)$. $\square$

Next, we write the difference $e^{\beta \cdot Q_h^k} - e^{\beta \cdot Q_h^*}$ in terms of $q_{h,1}^k$ and $q_{h,3}^k$.

**Lemma 5.** *For any $(k,h,s,a) \in [K] \times [H] \times \mathcal{S} \times \mathcal{A}$, suppose $(s,a)$ was previously visited at step $h$ of episodes $k_1, \ldots, k_t < k$. We have*

$$(e^{\beta \cdot Q_h^k} - e^{\beta \cdot Q_h^*})(s,a) = (q_{h,1}^k - q_{h,3}^k)(s,a).$$

*Proof.* For $e^{\beta \cdot Q_h^k}$, Line 10 of Algorithm 2 implies that

$$e^{\beta \cdot Q_h^k(s,a)} = q_{h,1}^k(s,a). \tag{41}$$

For $e^{\beta \cdot Q_h^*}$, we have from exponential Bellman equation (5) that

$$e^{\beta \cdot Q_h^*(s,a)} = e^{\beta \cdot r_h(s,a)} \left[ \mathbb{E}_{s' \sim P_h(\cdot \mid s,a)} e^{\beta \cdot V_{h+1}^*(s')} \right].$$

Let $t = N_h^k(s,a)$ and by Fact 1(d), we have

$$e^{\beta \cdot Q_h^*(s,a)} = \alpha_t^0 e^{\beta \cdot Q_h^*(s,a)} + \sum_{i \in [t]} \alpha_t^i e^{\beta \cdot r_h(s,a)} \left[ \mathbb{E}_{s' \sim P_h(\cdot \mid s,a)} e^{\beta \cdot V_{h+1}^*(s')} \right]$$

for each integer $t \geq 0$. By the definition of $q_{h,3}^k$ we have

$$e^{\beta \cdot Q_h^*(s,a)} = q_{h,3}^k(s,a). \tag{42}$$

The proof is completed by combining Equations (41) and (42). $\square$

From Lemma 5, we can derive the decomposition

$$(e^{\beta \cdot Q_h^k} - e^{\beta \cdot Q_h^*})(s,a) = (q_{h,1}^k - q_{h,2}^k)(s,a) + (q_{h,2}^k - q_{h,3}^k)(s,a) \tag{43}$$

if $\beta > 0$, and

$$(e^{\beta \cdot Q_h^k} - e^{\beta \cdot Q_h^*})(s,a) = (q_{h,2}^k - q_{h,1}^k)(s,a) + (q_{h,3}^k - q_{h,2}^k)(s,a) \tag{44}$$

if $\beta < 0$. We have the following lemmas.

**Lemma 6.** *There exists a universal constant $c > 0$ in the definition of $b_{h,t}$ in Algorithm 2 such that for any $(k, h, s, a) \in [K] \times [H] \times \mathcal{S} \times \mathcal{A}$ and $k_1, \ldots, k_t < k$ with $t = N_h^k(s, a)$, we have*

$$\left| \sum_{i \in [t]} \alpha_t^i \left[ e^{\beta[r_h(s,a) + V_{h+1}^*(s_{h+1}^{k_i})]} - \mathbb{E}_{s' \sim P_h(\cdot \mid s,a)} \left[ e^{\beta[r_h(s,a) + V_{h+1}^*(s')]} \right] \right] \right|$$

$$\leq c \left| e^{\beta(H-h+1)} - 1 \right| \sqrt{\frac{H\iota}{t}}.$$

*with probability at least $1 - \delta$, and*

$$\sum_{i \in [t]} \alpha_t^i b_{h,i} \in \left[ c \left| e^{\beta(H-h+1)} - 1 \right| \sqrt{\frac{H\iota}{t}}, 2c \left| e^{\beta(H-h+1)} - 1 \right| \sqrt{\frac{H\iota}{t}} \right].$$

*Proof.* We focus on the case where $\beta > 0$ and the proof for $\beta < 0$ is similar. For any $(k, h, s, a) \in [K] \times [H] \times \mathcal{S} \times \mathcal{A}$, define

$$\psi(i, k, h, s, a) := e^{\beta[r_h(s,a) + V_{h+1}^*(s_{h+1}^{k_i})]} - \mathbb{E}_{s' \sim P_h(\cdot \mid s,a)} \left[ e^{\beta[r_h(s,a) + V_{h+1}^*(s')]} \right]$$

$$= \mathbb{E}_{s' \sim \hat{P}_h^{k_i}(\cdot \mid s,a)} \left[ e^{\beta[r_h(s,a) + V_{h+1}^*(s')]} \right] - \mathbb{E}_{s' \sim P_h(\cdot \mid s,a)} \left[ e^{\beta[r_h(s,a) + V_{h+1}^*(s')]} \right]$$

Let us fix a tuple $(k, h, s, a) \in [K] \times [H] \times \mathcal{S} \times \mathcal{A}$. We have that $\{\mathbb{I}(k_i \leq K) \cdot \psi(i, k, h, s, a)\}_{i \in [\tau]}$ for $\tau \in [K]$ is a martingale difference sequence. By the Azuma-Hoeffding inequality and a union bound over $\tau \in [K]$, it holds that with probability at least $1 - \delta/(HSA)$, for all $\tau \in [K]$,

$$\left| \sum_{i \in [\tau]} \alpha_\tau^i \cdot \mathbb{I}(k_i \leq K) \cdot \psi(i, k, h, s, a) \right|$$

$$\leq \frac{c}{2}(e^{\beta(H-h+1)} - 1)\sqrt{\iota \sum_{i \in [\tau]} (\alpha_\tau^i)^2} \leq c(e^{\beta(H-h+1)} - 1)\sqrt{\frac{H\iota}{\tau}}$$

where $c > 0$ is some universal constant, the first step holds since $r_h(s, a) + V_{h+1}^*(s') \in [0, H - h + 1]$ for $s' \in \mathcal{S}$, and the last step follows from Fact 1(b). Since the above equation holds for all $\tau \in [K]$, it also holds for $\tau = t = N_h^k(s, a) \leq K$. Note that $\mathbb{I}(k_i \leq K) = 1$ for all $i \in [N_h^k(s, a)]$. Therefore, applying another union bound over $(h, s, a) \in [H] \times \mathcal{S} \times \mathcal{A}$, we have that the following holds for all $(k, h, s, a) \in [K] \times [H] \times \mathcal{S} \times \mathcal{A}$ and with probability at least $1 - \delta$:

$$\left| \sum_{i \in [t]} \alpha_\tau^i \cdot \psi(i, k, h, s, a) \right| \leq c(e^{\beta(H-h+1)} - 1)\sqrt{\frac{H\iota}{t}}, \tag{45}$$

where $t = N_h^k(s, a)$. Using the fact that $r_h + V_{h+1}^* \in [0, H - h + 1]$, we have

$$\left| \sum_{i \in [t]} \alpha_t^i \left[ \mathbb{E}_{s' \sim \hat{P}_h^{k_i}(\cdot \mid s,a)} e^{\beta[r_h(s,a) + V_{h+1}^*(s')]} - \mathbb{E}_{s' \sim P_h(\cdot \mid s,a)} e^{\beta[r_h(s,a) + V_{h+1}^*(s')]} \right] \right|$$

$$= \left| \sum_{i \in [t]} \alpha_t^i \cdot \psi(i, k, h, s, a) \right| \leq c(e^{\beta(H-h+1)} - 1)\sqrt{\frac{H\iota}{t}}.$$

For bounds on $\sum_{i \in [t]} \alpha_t^i b_{h,i}$, we recall the definition of $\{b_{h,t}\}$ in Line 8 of Algorithm 2 and compute

$$\sum_{i \in [t]} \alpha_t^i b_{h,i} = c(e^{\beta(H-h+1)} - 1) \sum_{i \in [t]} \alpha_t^i \sqrt{\frac{H\iota}{i}}$$

$$\in \left[ c(e^{\beta(H-h+1)} - 1)\sqrt{\frac{H\iota}{t}}, 2c(e^{\beta(H-h+1)} - 1)\sqrt{\frac{H\iota}{t}} \right]$$

where the last step holds by Fact 1(a). $\qquad \square$

The next two lemmas compare the iterate $e^{\beta \cdot Q_h^k}$ (and $e^{\beta \cdot V_h^k}$) with the optimal exponential value function $e^{\beta \cdot Q_h^*}$ (and $e^{\beta \cdot V_h^*}$).

**Lemma 7.** *For all $(k, h, s, a)$ and any $\delta \in (0, 1]$, it holds with probability at least $1 - \delta$ that*

$$\begin{cases} e^{\beta \cdot Q_h^k(s,a)} \geq e^{\beta \cdot Q_h^*(s,a)}, & \text{if } \beta > 0, \\ e^{\beta \cdot Q_h^k(s,a)} \leq e^{\beta \cdot Q_h^*(s,a)}, & \text{if } \beta < 0. \end{cases}$$

*Proof.* We focus on the case where $\beta > 0$ and the proof for $\beta < 0$ is similar. For the purpose of the proof, we set $Q_{H+1}^k(s, a) = Q_{H+1}^*(s, a) = 0$ for all $(k, s, a) \in [K] \times \mathcal{S} \times \mathcal{A}$. We fix a $(s, a) \in \mathcal{S} \times \mathcal{A}$ and use strong induction on $k$ and $h$. Without loss of generality, we assume that there exists a $(k, h)$ such that $(s, a) = (s_h^k, a_h^k)$ (that is, $(s, a)$ has been visited at some point in Algorithm 2), since otherwise $e^{\beta \cdot Q_h^k(s,a)} = e^{\beta(H-h+1)} \geq e^{\beta \cdot Q_h^*(s,a)}$ for all $(k, h) \in [K] \times [H]$ and we are done.

The base case for $k = 1$ and $h = H + 1$ is satisfied since $e^{\beta \cdot Q_{H+1}^{k'}(s,a)} = e^{\beta \cdot Q_{H+1}^*(s,a)}$ for $k' \in [K]$ by definition. We fix a $(k, h) \in [K] \times [H]$ and assume that $e^{\beta \cdot Q_{h+1}^{k_i}(s,a)} \geq e^{\beta \cdot Q_{h+1}^*(s,a)}$ for each $k_1, \ldots, k_t < k$ (here $t = N_h^k(s, a)$). Then we have for $i \in [t]$ that

$$e^{\beta \cdot V_{h+1}^{k_i}(s)} = \max_{a' \in \mathcal{A}} e^{\beta \cdot Q_{h+1}^{k_i}(s,a')} \geq \max_{a' \in \mathcal{A}} e^{\beta \cdot Q_{h+1}^*(s,a')} = e^{\beta \cdot V_{h+1}^*(s)},$$

where the first equality holds by the update procedure in Algorithm 2. Recall the decomposition in Equation (43). The above displayed equation implies $q_{h,1}^k \geq q_{h,2}^k$ by the definition of $q_{h,2}^k$. We also have $q_{h,2}^k \geq q_{h,3}^k$ by the fact $e^{\beta \cdot Q_h^*(s,a)} \leq e^{\beta(H-h+1)}$ and on the event of Lemma 6. Therefore, it follows that $(e^{\beta \cdot Q_h^k} - e^{\beta \cdot Q_h^*})(s, a) \geq 0$ by Equation (43). The induction is completed. $\qquad \square$

**Lemma 8.** *For all $(k, h, s, a) \in [K] \times [H] \times \mathcal{S} \times \mathcal{A}$ such that $t = N_h^k(s, a) \geq 1$, let $\gamma_{h,t} := 2 \sum_{i \in [t]} \alpha_t^i b_{h,i}$ and let $k_1, \ldots, k_t < k$ be the episodes in which $(s, a)$ is visited at step $h$. Then the following holds with probability at least $1 - \delta$: if $\beta > 0$, we have*

$$\begin{aligned} &(e^{\beta \cdot Q_h^k} - e^{\beta \cdot Q_h^*})(s, a) \\ &\leq \alpha_t^0 \left[ e^{\beta(H-h+1)} - 1 \right] + 2\gamma_{h,t} + \sum_{i \in [t]} \alpha_t^i e^{\beta} \left[ e^{\beta \cdot V_{h+1}^{k_i}(s_{h+1}^{k_i})} - e^{\beta \cdot V_{h+1}^*(s_{h+1}^{k_i})} \right], \end{aligned}$$

*and if $\beta < 0$, we have*

$$\begin{aligned} &(e^{\beta \cdot Q_h^*} - e^{\beta \cdot Q_h^k})(s, a) \\ &\leq \alpha_t^0 \left[ 1 - e^{\beta(H-h+1)} \right] + 2\gamma_{h,t} + \sum_{i \in [t]} \alpha_t^i \left[ e^{\beta \cdot V_{h+1}^*(s_{h+1}^{k_i})} - e^{\beta \cdot V_{h+1}^{k_i}(s_{h+1}^{k_i})} \right]. \end{aligned}$$

*Furthermore, we have $\gamma_{h,t} \leq 4c \left| e^{\beta(H-h+1)} - 1 \right| \sqrt{\frac{H\iota}{t}}$.*

*Proof.* Note that by definition,

$$q_{h,1}^k(s, a) = e^{\beta \cdot Q_h^k(s,a)}, \quad q_{h,3}^k(s, a) = e^{\beta \cdot Q_h^*(s,a)}.$$

Let us fix a tuple $(k, h, s, a) \in [K] \times [H] \times \mathcal{S} \times \mathcal{A}$. On the event of Lemma 7, we have

$$\begin{cases} e^{\beta \cdot Q_h^k(s,a)} \geq e^{\beta \cdot Q_h^*(s,a)}, & \text{if } \beta > 0, \\ e^{\beta \cdot Q_h^k(s,a)} \leq e^{\beta \cdot Q_h^*(s,a)}, & \text{if } \beta < 0. \end{cases}$$

This implies that for $i \in [t]$, if $\beta > 0$ then

$$e^{\beta \cdot V_{h+1}^{k_i}(s)} = \max_{a' \in \mathcal{A}} e^{\beta \cdot Q_{h+1}^{k_i}(s,a')} \geq \max_{a' \in \mathcal{A}} e^{\beta \cdot Q_{h+1}^*(s,a')} = e^{\beta \cdot V_{h+1}^*(s)},$$

and if $\beta < 0$, then

$$e^{\beta \cdot V_{h+1}^{k_i}(s)} = \min_{a' \in \mathcal{A}} e^{\beta \cdot Q_{h+1}^{k_i}(s,a')} \leq \min_{a' \in \mathcal{A}} e^{\beta \cdot Q_{h+1}^*(s,a')} = e^{\beta \cdot V_{h+1}^*(s)}.$$

Here, the first equalities for the above two displays follow from the update procedure in Algorithm 2.

**Case $\beta > 0$.** We have

$$
(q_{h,1}^k - q_{h,2}^k)(s,a) \overset{(i)}{\leq} (q_{h,1}^{k,+} - q_{h,2}^{k,\circ})(s,a)
$$

$$
\overset{(ii)}{\leq} \sum_{i \in [t]} \alpha_t^i \left[ e^{\beta[r_h(s,a)+V_{h+1}^{k_i}(s_{h+1}^{k_i})]} - e^{\beta[r_h(s,a)+V_{h+1}^*(s_{h+1}^{k_i})]} \right] + \sum_{i \in [t]} \alpha_t^i b_{h,i}
$$

$$
\leq \sum_{i \in [t]} \alpha_t^i \cdot e^{\beta} \left[ e^{\beta \cdot V_{h+1}^{k_i}(s_{h+1}^{k_i})} - e^{\beta \cdot V_{h+1}^*(s_{h+1}^{k_i})} \right] + \gamma_{h,t}
$$

where step $(i)$ holds by the fact that $\alpha_t^0, \sum_{i \in [t]} \alpha_t^i \in \{0,1\}$ with $\alpha_t^0 + \sum_{i \in [t]} \alpha_t^i = 1$ by Fact 1(d) (so that $q_{h,1}^k \geq q_{h,2}^k$); step $(ii)$ holds by definitions of $q_{h,1}^{k,+}$ and $q_{h,2}^{k,\circ}$; the last step holds since $r_h$ is in $[0,1]$ entrywise and $V_{h+1}^{k_i}(s) \geq V_{h+1}^*(s)$. Moreover, we have

$$
(q_{h,2}^k - q_{h,3}^k)(s,a) \overset{(i)}{\leq} (q_{h,2}^{k,+} - q_{h,3}^k)(s,a)
$$

$$
= \alpha_t^0 \left[ e^{\beta(H-h+1)} - e^{\beta \cdot Q_h^*(s,a)} \right] + \sum_{i \in [t]} \alpha_t^i b_{h,i}
$$

$$
+ \sum_{i \in [t]} \alpha_t^i \left[ e^{\beta[r_h(s,a)+V_{h+1}^*(s_{h+1}^{k_i})]} - \mathbb{E}_{s' \sim P_h(\cdot \mid s,a)}[e^{\beta[r_h(s,a)+V_{h+1}^*(s')]}] \right]
$$

$$
\leq \alpha_t^0 \left[ e^{\beta(H-h+1)} - 1 \right] + \gamma_{h,t},
$$

where step $(i)$ holds by

$$
\sum_{i \in [t]} \alpha_t^i b_{h,i} \geq \left| \sum_{i \in [t]} \alpha_t^i \left[ e^{\beta[r_h(s,a)+V_{h+1}^*(s_{h+1}^{k_i})]} - \mathbb{E}_{s' \sim P_h(\cdot \mid s,a)}[e^{\beta[r_h(s,a)+V_{h+1}^*(s')]}] \right] \right|
$$

on the event of Lemma 6 (so that $q_{h,2}^k \geq q_{h,3}^k$) and Fact 2; the last step holds by $Q_h^* \geq 0$ and on the event of Lemma 6.

**Case $\beta < 0$.** We have

$$
(q_{h,2}^k - q_{h,1}^k)(s,a) \overset{(i)}{\leq} (q_{h,2}^{k,\circ} - q_{h,1}^{k,+})(s,a)
$$

$$
= \sum_{i \in [t]} \alpha_t^i \left[ e^{\beta[r_h(s,a)+V_{h+1}^*(s_{h+1}^{k_i})]} - e^{\beta[r_h(s,a)+V_{h+1}^{k_i}(s_{h+1}^{k_i})]} \right] + \sum_{i \in [t]} \alpha_t^i b_i
$$

$$
\leq \sum_{i \in [t]} \alpha_t^i \left[ e^{\beta \cdot V_{h+1}^*(s_{h+1}^{k_i})} - e^{\beta \cdot V_{h+1}^{k_i}(s_{h+1}^{k_i})} \right] + \gamma_{h,t},
$$

where the step $(i)$ holds since $q_{h,2}^{k,\circ} \geq q_{h,2}^k$ by Fact 2 and $q_{h,1}^{k,+} \leq q_{h,1}^k$ by definition, and the last step holds by the fact that $r_h(s,a) + V_{h+1}^{k_i}(s) \geq r_h(s,a) + V_{h+1}^*(s)$, that $e^{\beta \cdot r_h(s,a)} \leq 1$ given $\beta < 0$, and the definition of $\gamma_{h,t}$. In addition, we can derive

$$
(q_{h,3}^k - q_{h,2}^k)(s,a) \overset{(i)}{\leq} (q_{h,3}^k - q_{h,2}^{k,+})(s,a)
$$

$$
= \alpha_t^0 \left[ 1 - e^{\beta \cdot Q_h^*(s,a)} \right] + \sum_{i \in [t]} \alpha_t^i b_i
$$

$$
+ \sum_{i \in [t]} \alpha_t^i \left[ \mathbb{E}_{s' \sim P_h(\cdot \mid s,a)}[e^{\beta[r_h(s,a)+V_{h+1}^*(s')]}] - e^{\beta[r_h(s,a)+V_{h+1}^*(s_{h+1}^{k_i})]} \right]
$$

$$
\overset{(ii)}{\leq} \alpha_t^0 \left[ 1 - e^{\beta(H-h+1)} \right] + 2 \sum_{i \in [t]} \alpha_t^i b_i
$$

$$\leq \alpha_t^0 \left[ 1 - e^{\beta(H-h+1)} \right] + \gamma_{h,t}.$$

where step $(i)$ holds since $q_{h,2}^k \geq q_{h,2}^{k,+}$, step $(ii)$ holds on the event of Lemma 6, and the last step holds by the definition of $\gamma_{h,t}$.

Combining the above calculations with Equation (43) for the case where $\beta > 0$ (or Equation (44) for the case where $\beta < 0$) yields the upper bound for $(e^{\beta \cdot Q_h^k} - e^{\beta \cdot Q_h^*})(s,a)$ (or $(e^{\beta \cdot Q_h^*} - e^{\beta \cdot Q_h^k})(s,a)$). Furthermore, Lemma 6 and the definition of $\gamma_{h,t}$ together imply

$$\gamma_{h,t} \leq 4c \left| e^{\beta(H-h+1)} - 1 \right| \sqrt{\frac{H\iota}{t}}.$$

The proof is completed. $\qquad\qquad\qquad\qquad\qquad\qquad\qquad\qquad\qquad\qquad\qquad\qquad\square$

We present a simple inequality for the regret.

**Lemma 9.** *Suppose that for any $k \in [K]$ we have $V_1^k(s_1^k) \geq V_1^*(s_1^k)$. Then for $\beta > 0$, the regret is bounded by*

$$\mathrm{Regret}(K) \leq \frac{1}{\beta} \sum_{k \in [K]} [e^{\beta \cdot V_1^k(s_1^k)} - e^{\beta \cdot V_1^{\pi^k}(s_1^k)}],$$

*and for $\beta < 0$, the regret is bounded by*

$$\mathrm{Regret}(K) \leq \frac{e^{-\beta H}}{|\beta|} \sum_{k \in [K]} [e^{\beta \cdot V_1^{\pi^k}(s_1^k)} - e^{\beta \cdot V_1^k(s_1^k)}],$$

*Proof.* For $\beta > 0$, we have

$$\mathrm{Regret}(K) = \sum_{k \in [K]} (V_1^* - V_1^{\pi^k})(s_1^k)$$

$$\overset{(i)}{\leq} \sum_{k \in [K]} (V_1^k - V_1^{\pi^k})(s_1^k)$$

$$= \sum_{k \in [K]} \left[ \frac{1}{\beta} \log\{e^{\beta \cdot V_1^k(s_1^k)}\} - \frac{1}{\beta} \log\{e^{\beta \cdot V_1^{\pi^k}(s_1^k)}\} \right]$$

$$\overset{(ii)}{\leq} \sum_{k \in [K]} \frac{1}{\beta} [e^{\beta \cdot V_1^k(s_1^k)} - e^{\beta \cdot V_1^{\pi^k}(s_1^k)}]$$

$$= \frac{1}{\beta} \sum_{k \in [K]} [e^{\beta \cdot V_1^k(s_1^k)} - e^{\beta \cdot V_1^{\pi^k}(s_1^k)}],$$

where step $(i)$ holds by our assumption, and step $(ii)$ holds by the 1-Lipschitzness of the function $f(x) = \log x$ for $x \geq 1$ and note that our assumption implies that $V_1^k(s_1^k) \geq V_1^*(s_1^k) \geq V_1^{\pi^k}(s_1^k)$.

For $\beta < 0$, we similarly have

$$\mathrm{Regret}(K) = \sum_{k \in [K]} (V_1^* - V_1^{\pi^k})(s_1^k)$$

$$\overset{(i)}{\leq} \sum_{k \in [K]} (V_1^k - V_1^{\pi^k})(s_1^k)$$

$$= \sum_{k \in [K]} \left[ \frac{1}{\beta} \log\{e^{\beta \cdot V_1^k(s_1^k)}\} - \frac{1}{\beta} \log\{e^{\beta \cdot V_1^{\pi^k}(s_1^k)}\} \right]$$

$$= \sum_{k \in [K]} \left[ \frac{1}{(-\beta)} \log\{e^{\beta \cdot V_1^{\pi^k}(s_1^k)}\} - \frac{1}{(-\beta)} \log\{e^{\beta \cdot V_1^k(s_1^k)}\} \right]$$

$$\overset{(ii)}{\leq} \sum_{k \in [K]} \frac{e^{-\beta H}}{(-\beta)} [e^{\beta \cdot V_1^{\pi^k}(s_1^k)} - e^{\beta \cdot V_1^k(s_1^k)}]$$

$$= \frac{e^{-\beta H}}{|\beta|} \sum_{k \in [K]} [e^{\beta \cdot V_1^{\pi^k}(s_1^k)} - e^{\beta \cdot V_1^k(s_1^k)}],$$

where step $(i)$ holds by our assumption, and step $(ii)$ holds by the $(e^{-\beta H})$-Lipschitzness of the function $f(x) = \log x$ for $x \geq e^{\beta H}$ and note that our assumption implies that $V_1^k(s_1^k) \geq V_1^*(s_1^k) \geq V_1^{\pi^k}(s_1^k)$. $\qquad \square$

**Broader impact and future directions.** Risk-sensitive RL has close association with neuroscience, psychology and behavioral economics, as it has been applied to model human behaviors [36, 41]. Interestingly, this array of topics are also actively studied by researchers in the areas of meta learning [44], biologically inspired deep learning [42] and deep reinforcement learning [29]. It would be an exciting research direction to establish connections between these related areas through rigorous and theoretical analysis of deep learning [9, 11]. Motivated by the inertia of switching actions that is widely observed in human behaviors, the study of switching constrained algorithms [12] for risk-sensitive RL could be another promising direction for future investigation. Furthermore, to make our algorithms practical and efficient on large-scaled datasets collected in the aforementioned applications, it is imperative to enable offline learning procedures for risk-sensitive RL, possibly by techniques developed in the literature of offline RL [10]. It would also be of great interest to understand the landscape of the optimization problems [30] that arise in the offline learning setting.