# OpenReview forum: "Exponential Bellman Equation and Improved Regret Bounds for Risk-Sensitive Reinforcement Learning"
_NeurIPS.cc/2021/Conference — NeurIPS 2021 Poster_

### Official Review · Reviewer_nUcE · 2021-07-01

**Rating:** 6
**Confidence:** 3

**Summary:**

The paper studies the exponential Bellman equation that arises in risk sensitive reinforcement learning when using entropic risk measure as the risk measure. It claims to "offer a route for connecting distributional RL and risk sensitive RL", although this question is never seriously addressed. The main result of their analysis consists in design and analyse two novel RL algorithms that employ a novel exploration mechanism. The regret bounds on this mechanism improves by an exponential factor.

**Limitations And Societal Impact:**

I do not see anything special to report here.

**Main Review:**


Generally, the paper is well written and tackles an interesting problem. I consider the main contribution to be the said algorithms that exploit a new exploration strategy and its analysis in terms of regret, which leads to a regret bound that is O(exp(H)) instead of O(exp(H^2)) for previously proposed algorithm for this problem. From a theoretical point of view, this seems to be a relatively important contribution. On the other hand, I find that the claimed second contribution which relates to distributional RL is rather under-developed and trivial in its current state. The paper also might also lack a concrete application where one can confirm that the theoretical performance improvement translates in actual speed-ups in convergence rate during training.

Regarding the connection with distributional RL, the authors claim in the abstract and introduction that the paper answers to the need of identifying an algorithm that can optimize non-heuristically risk-sensitive policies in a distributional RL framework. Yet, the only support to this claim comes in the form of an observation that the use of a recursive entropic risk measure in distributional RL might motivate encoding the distribution of the "total cost to go" through its moment generating function. Implications of this observation are completely missing from the paper while the details of a rigorous algorithmic method to do risk-sensitive distributional RL are undisclosed.


Other comments:

Why focus on finite horizon MDPs? This assumption significantly restricts the scope of the paper.

Equation (4) is misleading as it seems to indicate that the same Q_h function needs to satisfy the equality constraint for all values of \mu. In general, I suspect that this is impossible to satisfy and that the authors meant that given any fixed \mu, one could impose the condition described in equation (4). In particular, in a distributional RL setting, my understanding is that the Q_pi function would need to have four input parameters: h, s, a, \mu.


**Time Spent Reviewing:**

2

---

> ### Author Response · Authors · 2021-08-10
> **Response to Reviewer nUcE**
>
> Thank you for your positive comments. We would like to address your concerns as follows.
>
>
> - “second contribution which relates to distributional RL”
>
>    In distributional RL, one aims to estimate the value distribution, or in other words, the distribution of cumulative rewards. Inspired by the exponential Bellman equation (4), we may estimate the MGF of cumulative rewards. Note that an MGF uniquely identifies a distribution, just like quantiles or distribution functions used in existing distributional RL algorithms. To that end, in theory, one may obtain an estimate of the MGF of the reward distribution by running our Algorithm 1 or 2 for all $\beta \ne 0$. In practice, one may run the algorithms for values of $\beta$ within a predetermined grid. The choice of grid depends on the available computational power, prior knowledge or estimate of the heavy-tailedness of the reward distribution, and etc. To the best of our knowledge, this is the first time that a connection is made between distributional RL and risk-sensitive RL with entropic risk measure. We will add this clarification to the final version.
>
>
>
>
> - “concrete application”
>
>    Our paper focuses on theoretical aspects of risk-sensitive RL. It is a great suggestion to conduct numerical experiments to support our theoretical results. We will follow up on this.
>
>
> - “Why focus on finite horizon MDPs”
>
>    We focus on the finite-horizon setting because it is the most common setup for studying sample complexity of RL algorithms. Our algorithm and theory can be extended to infinite-horizon discounted and average-reward settings, which we leave as future work.
>
>
> - “Equation (4) is misleading”
>
>    In Eq (4), the Q function does implicitly depend on $\mu$, and we mean to say that the Q function (implicitly parameterized by $\mu$) satisfies Eq (4) for each fixed $\mu$. We will add this point to the final version of the paper.

---

> > ### Comment · Area_Chair_SF43 · 2021-08-26
> > **Distributional RL Claims**
> >
> > Despite the author response, Reviewer nUcE's initial review concerns about the mostly undeveloped link to Distributional RL (also reflected in other reviews) still appear to be quite valid.  This seems problematic because even the paper title alludes to Distributional RL.  Would it be a mistake to completely remove the Distributional RL claims from the title and the paper?

---

> > > ### Author Response · Authors · 2021-08-26
> > > **Response to "Distributional RL Claims"**
> > >
> > > Dear Area Chair,
> > >
> > > Thank you for your comment. We would like to thank Reviewer nUcE's comment regarding the connection between distributional RL and risk-sensitive RL.  In the revised version, we will  _remove the connection to distributional RL from our main contributions_ and only discuss such a connection in the paper as a side remark.
> > >
> > > We would like to highlight that our theoretical results are **valid** even if we completely neglect the distributional RL part. The reason is our algorithms leverage the _exponential Bellman equation_, whose solution yields the policy that maximizes the _entropic risk measure_. In our algorithms, we  propose a novel analysis of the Bellman backup procedure and a novel bonus design named as doubly decaying bonus. These novel ingredients enable us to achieve **sample efficiency** in risk-sensitive RL.
> > >
> > > Risk-sensitive RL is an important direction in decision science and the study of risk-sensitive RL via the lens of entropic risk measure (also known as exponential utility function) dates back to the '70s (See, e.g., [1],[2],[3]). In particular, [2] extends the classical Q-learning algorithm to entropic risk measure and establish asymptotic convergence via the techniques of differential equations. The Q-learning algorithm established in our work brings significant improvement in that our algorithm efficiently balances the _exploration-exploitation tradeoff_ for the entropic risk measure. Compared with the latest literature on risk-sensitive RL, in the paper, we have shown that our algorithms achieve _nearly optimal regret bounds_ which improve upon the existing literature. We believe our work brings **solid algorithmic and theoretical contribution** to the **literature of risk-sensitive RL**. Neglecting the connection to distributional RL does not affect this part at all.
> > >
> > > In the revised version, we will _remove the connection to distributional RL from our contributions_ and only discuss such a connection in the paper as a side remark. In particular, we will explain in detail that the connection between Distributional RL and Risk-sensitive RL exists in the following two aspects.
> > >
> > > First, our objective, the entropic risk measure, when viewed as a function of $\beta$, corresponds to the _moment generating function_ (MGF) of the cumulative reward. Thus, our work can be used to estimate the MGF by choosing a grid of $\beta$'s. Once we find the MGF, we recover the distribution of the cumulative reward, which is the central object in distributional RL. This view is analogous to popular distributional RL algorithms based on _quantile functions_ (e.g., [4]). In particular, in [4], the authors proposes to estimate the quantile function $F_{\tau} $ for a grid of quantile parameter $\tau$. Quantile function is another characterization of the distribution, similar to MGF, while our $\beta$ is similar to their $\tau$. However, there is **no Bellman equation associated with the quantile function**, which makes it hard to estimate with theoretical guarantees.  In sharp contrast, for MGF, there exists a Bellman equation, namely, the exponential Bellman equation, which builds the theoretical foundation for characterizing the distribution of cumulative rewards via MGFs.
> > >
> > > Second, in distributional RL, it is often preferable to have a risk-sensitive policy. For example, [4] shows empirically that _risk-averse version of IQN achieves better performances_. In [4], risk sensitivity is achieved via a weighted integration over quantiles. Whereas our entropic risk measure directly encodes risk sensitivity in parameter $\beta$. While developing a distributional RL algorithm is out of the scope of our work, our focus is to utilize a distributional perspective to solve the risk-sensitive RL problem, which is explored empirically by [4] via quantile networks. Similar to quantiles, our entropic risk measure satisfies two ideal properties: (i) being able to characterize the distribution of the cumulative rewards, and (ii) achieve risk sensitivity.
> > >
> > >
> > > -------------------------------------------
> > > **Reference**
> > >
> > > [1] Ronald A. Howard, James E. Matheson, (1972) Risk-Sensitive Markov Decision Processes. Management Science 18(7):356-369.
> > > https://doi.org/10.1287/mnsc.18.7.356
> > >
> > > [2] V. S. Borkar, (2002) Q-Learning for Risk-Sensitive Control. Mathematics of Operations Research 27(2):294-311. https://
> > > doi.org/10.1287/moor.27.2.294.324
> > >
> > > [3] P. Whittle, (1981) Risk-Sensitive Linear/Quadratic/Gaussian Control. Advances in Applied Probability
> > > Vol. 13, No. 4 (Dec., 1981), pp. 764-777 (14 pages)
> > >
> > > [4] Dabney et al (2018)  Implicit Quantile Networks for Distributional Reinforcement Learning. ICML

---

> > > > ### Comment · Reviewer_tvfS · 2021-08-27
> > > > **Clarification about MGF of cumulative rewards**
> > > >
> > > > As far as I understand, the MGF of the cumulative rewards would be written as:
> > > > \mathbb E_Z[ e^{\mu Z} ]
> > > > which seems to be different from (4) where the expectation is only taken over future states (the distribution over future rewards has been already eliminated).
> > > > Could you clarify that point?

---

> > > > > ### Author Response · Authors · 2021-08-27
> > > > > **Re: Clarification about MGF of cumulative rewards**
> > > > >
> > > > > Dear reviewer,
> > > > >
> > > > > Thank you for your question.
> > > > >
> > > > > In Eq.(2) we introduce the distributional Bellman equation
> > > > > $$
> > > > > Z_h^\pi(s,a) \stackrel{dist.}{=} r_h(s,a) + Z_{h+1}^{\pi}(X', U') ,\qquad \forall h\in [H] ,
> > > > > $$
> > > > > where $Z_h^\pi (s,a)$ is the random variable that is equal to the _random cummulative rewards_ under policy $\pi$ at step  $h$ and state-action $(s,a)$. Here $X' $ is the (random) next state and $U' \sim \pi(\cdot | X') $ is the random action sampled from policy $\pi$, and $\stackrel{dist.}{=}$ means having the same distribution.
> > > > >
> > > > > Meanwhile, in Eq.(4) we introduce the exponential Bellman equation, which is recursively defined by
> > > > > \begin{align}
> > > > > Q_h^\pi (s,a) & = \frac{1}{\mu} \log \bigg  ( \mathbb{E} _{X' \sim P(\cdot | s,a) } \Big [ \exp \Big(  \mu \cdot  \big ( r_h(s,a) + V _{h+1} ^{\pi}(X')      \big)  \Big ) \Big]  \bigg) ,
> > > > > \end{align}
> > > > > \begin{align}
> > > > > V_h^{\pi} (s) & =  Q_h^\pi (s, \pi(s) )  .
> > > > > \end{align}
> > > > > Here we overload the definition of state- and action-value functions and the definitions are the same as the classical ones when $\mu = 0$ (risk-neutral). Moreover, in $Q_h^\pi (s, \pi(s) ) $ we also overload the notation, which means that in the definition of $Q^{\pi}(s,a)$ we plug in $a = \pi(s)$. This makes sense because we focus on _deterministic_ policies. More generally, when $\pi$ is a stochastic policy, we define $V_h^\pi$ as
> > > > > \begin{align}
> > > > > V_h^\pi(s) =  \frac{1}{\mu} \log \bigg  ( \mathbb{E} _{a\sim \pi(\cdot|s), X' \sim P(\cdot | s,a) } \Big [ \exp \Big(  \mu \cdot  \big ( r_h(s,a) + V _{h+1} ^{\pi}(X')      \big)  \Big ) \Big]  \bigg) .
> > > > > \end{align}
> > > > > This reduces to $ Q_h^\pi (s, \pi(s) ) $ when $\pi$ is deterministic.
> > > > >
> > > > > Thus, by unrolling the Bellman equation, we further have (also see Eq. (6) in the paper)
> > > > > \begin{align}
> > > > > V_h^{\pi} (s) = \frac{1}{\mu} \log \bigg  ( \mathbb{E}   \Big [ \exp \Big(  \mu  \cdot   {\textstyle \sum}_{i=h}^H r_i (s_i, a_i)         \Big ) \Big | s_h = s, a_j \sim \pi(\cdot | s_j ), s _ {j+1}  \sim P(\cdot | s_j,  a_j) , \forall j \geq h \Big]  \bigg) .
> > > > > \end{align}
> > > > >
> > > > > \begin{align}
> > > > > Q_h^{\pi} (s, a ) = \frac{1}{\mu} \log \bigg  ( \mathbb{E}   \Big [ \exp \Big(  \mu  \cdot   {\textstyle \sum}_{i=h}^H r_i (s_i, a_i)         \Big ) \Big | s_h = s, a_h = a , a_j \sim \pi(\cdot | s_j ), s _ {j+1}  \sim P(\cdot | s_j,  a_j) , \forall j \geq h + 1 \Big]  \bigg) .
> > > > > \end{align}
> > > > >
> > > > > Now notice that given $(s_h , a_h ) = (s,a)$ and $a_j \sim \pi(\cdot | s_j) , \forall j \geq h+1$,
> > > > > $ \sum _{i=h}^H r_i (s_i, a_i) $ is the (random) cummulative reward starting from the $h$-th timestep. That is,
> > > > > \begin{align}
> > > > > Z_h^\pi(s,a) \stackrel{dist.}{=}\sum _{i=h}^H r_i (s_i, a_i).
> > > > > \end{align}
> > > > >  Therefore, we have
> > > > > \begin{align}
> > > > > Q_h^{\pi} (s, a ) = \frac{1}{\mu} \log \bigg  ( \mathbb{E}   \Big [ \exp \Big(\mu \cdot  Z_h^\pi(s,a)   \Big]  \bigg) = \frac{1}{\mu} \log \bigg ( \mathtt{MGF} _ {Z_h^\pi(s,a)}  (\mu)   \bigg)  ,
> > > > > \end{align}
> > > > > where we let $\mathtt{MGF} _  Z (\cdot )$ denote the moment-generating function of any random variable $Z$, i.e., $\mathtt{MGF} _  Z (t ) = \mathbb{E} [ \exp(t \cdot Z) ]$.
> > > > >
> > > > > We thank the reviewer for the question. We will better present this connection in the revised version.

---

### Official Review · Reviewer_tvfS · 2021-07-16

**Rating:** 7
**Confidence:** 3

**Summary:**

The paper considers risk-sensive decision-making in reinforcement learning as modeled by an exponential utility function. In contrast to a previous work by Fei et al. (2016), the authors propose to consider the exponential Bellman equations, which allows to provide a smaller regret upperbound of their proposed algorithms.

**Limitations And Societal Impact:**

No suggestion is needed since the work is more fundamental.

**Main Review:**

The paper is generally well-written and the contributions (Algorithms with doubly decaying bonus and theoretical analysis that reduces the gap between the upperbound and lowerbound) are clearly presented. However, I think the exposition would be clearer if Sec.3.3 happens before Sec.3.2 because the exponential Bellman equation makes sense only after the corresponding value function has been introduced.

Regarding the related work about distributional RL, the authors may want to also cite the following paper:
D. Yang, L. Zhao, Z. Lin, T. Qin, J. Bian, and T. Liu, “Fully Parameterized Quantile Function for Distributional Reinforcement Learning,” NeurIPS 2019

Typos:
- line 96: large -> larger
- (2): Z_{H+1} = 0 should be added
- (3): Z_{h+1} -> Z
- line 217: differerent
- Algorithm 1 should also include the case when \beta < 0 in lines 8 an 11
- line 286: aveage
- line 288: miminial
- line 306: Algorithm -> Algorithms

**Time Spent Reviewing:**

4

---

> ### Author Response · Authors · 2021-08-10
> **Response to Reviewer tvfS**
>
> Thank you for your positive comments. We would like to address your concerns as follows.
>
>
> - “Sec.3.3 happens before Sec.3.2”
>
>    We will improve on the exposition regarding this point.
>
>
> - “Regarding the related work about distributional RL”
>
>    Thank you for pointing out the additional reference on distributional RL. We will cite the paper in our final version.
>
>
> We appreciate that you point out the typos. We will correct them in the final version of the paper.

---

### Official Review · Reviewer_YKaB · 2021-07-16

**Rating:** 6
**Confidence:** 3

**Summary:**

In this work a theoretical analysis is conducted in the context of risk-sensitive reinforcement learning with entropic risk measures. This allows connecting distributional RL and risk-sensitive RL. Two algorithms are proposed with a novel exploration scheme based on two bonuses. Regrets bounds are provided that improve previous analyses by an exponential term.


**Limitations And Societal Impact:**

I think the authors could have done a better job stating the limitations of their work.

**Main Review:**

Originality:

The paper relies heavily on  a previous paper [16]. However, the authors improve [16] by a simple procedure re-formulating a recursion using an exponential form. The result of this re-formulation gives insight on how to obtain a tighter bound which translate in very simple modifications to existent algorithms (RSVI and RSQ from [16]). Although the change is small it has considerable implications in the regret bound.

I also think the literature is fairly represented, specially the section commenting on the paper [16] since it is the closest one to this work.

Quality:

- I believe the work is technically sound.

- I am unsure about the connection to distributional RL claimed by the authors. Maybe they could elaborate more on that? As far as I can see Equation 4 is just a simple re-arrangement of Equation 8 which is very well known in the community (see N Tishby, PA Ortega, J Grau-Moya, R Fox, D Polani works to name a few) and also derived in [16]. How exactly this connects to distributional RL?

- It would be interesting to see if the authors can comment how their work could be extended to infinite horizon settings or to the online setting (non-episodic). Since the algorithm requires revisiting states at time "h" in order to have good estimates of Equation 5 I wonder if the presented work breaks down in this case.

- Shouldn't Line 8 of Algorithm 1 contain a log e.g. 1/beta log G? Hand-wavy explanation: Since V=max Q and exp(beta Q) = G--> V = max 1/beta log G?

- Actually looking again at this line, the authors could comment what happens when there is only 1 sample to estimate w? In that case one would obtain a risk-neutral estimate of V right?

Clarity:

I want to thank the authors for producing such a well explained work. It is well organized, the explanations of the problems in [16] were good, and the method the authors used to solve it was also very well explained.

Significance:

I think since the bound improvements are fairly easy to re-construct, this work might be a reference  that future researchers might use.



**Time Spent Reviewing:**

4

---

> ### Author Response · Authors · 2021-08-10
> **Response to Reviewer YKaB**
>
> Thank you for your positive comments. We would like to address your concerns as follows.
>
> - “connection to distributional RL”
>
>    In distributional RL, one aims to estimate the value distribution, or in other words, the distribution of cumulative rewards. Inspired by the exponential Bellman equation (4), we may estimate the MGF of cumulative rewards. Note that an MGF uniquely identifies a distribution, just like quantiles or distribution functions used in existing distributional RL algorithms. To that end, in theory, one may obtain an estimate of the MGF of the reward distribution by running our Algorithm 1 or 2 for all $\beta \ne 0$. In practice, one may run the algorithms for values of $\beta$ within a predetermined grid. The choice of grid depends on the available computational power, prior knowledge or estimate of the heavy-tailedness of the reward distribution, and etc. To the best of our knowledge, this is the first time that a connection is made between distributional RL and risk-sensitive RL with entropic risk measure. We will add this clarification to the final version.
>
>
> - “extended to infinite horizon settings or to the online setting (non-episodic)”
>
>    A possible direction for extending our work to infinite horizon and non-episodic settings is to follow the reduction to finite horizon proposed by, e.g., [1*] (discounted setting) and [2*, 3*] (average reward). For example, the discounted setting entails an effective horizon of $1/(1-\gamma)$. More specifically, upon a truncation at the $1/(1-\gamma)$-th step, the discounted setting reduces to a finite horizon at the cost of a diminishing bias. However, we would like to point out a technical difference between the regret defined in [1*] and the regret defined in a few other papers, e.g., [4*]. It remains unclear how to obtain the latter notion through such a reduction. We believe it is an interesting and important future research direction.
>
>
>
>
>
>
> - “Shouldn't Line 8 of Algorithm 1 contain a log e.g. 1/beta log G”
>
>    Yes you are certainly correct and thanks for picking out the typo. We will correct it in the final version.
>
>
> - “what happens when there is only 1 sample to estimate w”
>
>    Given only one sample, our algorithms still estimate the risk-sensitive value function. For example, one may set $N_h(s,a) = 1$ in Line 5 of Algorithm 1 and use the resulting $w_h(s,a)$ in Line 7. It can be seen that $G_h(s,a)$ still estimates the exponentiated value function, and therefore $V_h(s)$ in Line 9 is a risk-sensitive estimate.
>
>
>
> - “Limitations”
>
>    One limitation of our work is that our results are restricted to the finite-horizon and tabular setting. It would be interesting to consider extensions to infinite-horizon or function approximation settings. Besides, there exists a gap between our regret upper bound and the existing regret lower bound which is exponential in horizon. Closing the gap might require more sophisticated algorithm design or regret analysis, and we leave that as future work.
>
>
> [1*] Zhou et al (2021). Provably Efficient Reinforcement Learning for Discounted MDPs with Feature Mapping.
>
>
> [2*] Wei et al (2020). Model-free Reinforcement Learning in Infinite-horizon Average-reward Markov Decision Processes.
>
>
> [3*] Wei et al (2021). Learning Infinite-horizon Average-reward MDPs with Linear Function Approximation.
>
>
> [4*] Dong et al (2019). Q-learning with UCB Exploration is Sample Efficient for Infinite-Horizon MDP.

---

> > ### Comment · Reviewer_YKaB · 2021-08-31
> > **clarification**
> >
> > Thanks for your reply.
> >
> > With respect to the "1-sample" question: Are you sure this is correct? See below the rationale (using simple notation but referring to the quantities in the paper)
> > If we use Line 5 with 1 sample we have $w=e^{\beta(r+V)}$
> > Then we go to Line 7 (disregarding $b$ and the $\min$ operation) we have $G = \frac{1}{\beta} \log w = r + V$ which is just the standard risk-neutral Q-value function.
> >
> > edit: Equation --> Line

---

> > > ### Author Response · Authors · 2021-08-31
> > > **Re clarification**
> > >
> > > Thank you for your question.
> > >
> > > We agree with you that if one disregards the bonus $b$, then the quantity $G$ in Line 7 indeed estimates the risk-neutral Q-function in the one-sample setting. On the other hand, we do want to point out that our previous answer pertains to the case where we keep the bonus term in Line 7. This is because the bonus term is a crucial component of our algorithms and should not be ignored. On the algorithm side, the bonus enables the algorithms to explore the state-action space in the uncertain environment; on the theoretical side, the bonus is the key factor that allows us to derive sub-linear regret bounds. In particular, our improved regret bounds crucially rely on the novel and careful design of the bonus, which we call the *doubly decaying* bonus (please see Section 5.2 for details on the topic).

---

> > > > ### Comment · Reviewer_YKaB · 2021-08-31
> > > > **re**
> > > >
> > > > Thanks for the quick reply.
> > > >
> > > > In my view the bonus controls exploration taking into account the horizon and visitations, whereas the 1/beta log sum exp is the entropic risk measure that really accounts for risk-sensitivity. *Correct me if I am wrong but even if you keep the bonus the $w$-estimate with 1 sample is going to be risk-neutral right?* I say this because there is no way to capture anything related to the distribution of the return via the terms inside the bonus $b$. Here I not attacking the methodology that you are employing, it is just a curiosity from my side that requires some clarification from the experts (you), because I couldn't make much sense of the original response.

---

> > > > > ### Author Response · Authors · 2021-08-31
> > > > > **Re clarification**
> > > > >
> > > > > Yes, your understanding is correct. More formally, we should say that $G$ is an *optimistic* estimate of the risk-sensitive (exponentiated) Q-fucntion. When we use one sample in Line 5 and leave out the bonus and thresholding operation in Line 7, the iterate $V$ in Line 8 coincides with the one-sample estimate of the risk-neutral value function.
> > > > >
> > > > > To further illustrate our point, consider the "1-sample" case, neglecting the $b$ term, indeed we have $w (s,a) = \exp( \beta \cdot (r + V) ) $ if $(s,a) = (s^1, a^1)$ and $ w (s,a) = \exp(H - h)$ otherwise. Here $(s^1, a^1)$ is the data point. Now, for $(s,a) = (s^1, a^1)$, indeed the estimator, $\log w$, becomes $r  + V$, and on other points, i.e., $(s,a) \neq (s^1 , a^1)$, our estimator is $H - h$.
> > > > >
> > > > > Here the **risk-sensitive** estimator coincides with the **risk-neutral** counterpart only because we **only have one sample**. These two estimators becomes very different when we collect more and more data.
> > > > >
> > > > > One might think that, when we have only one sample, we estimate the risk-neutral value function instead of the desired exponentiated value function, and thus the estimation error is large. This is indeed correct. But more importantly, the estimation uncertainty is _dominated by the bonus function $b$_, which enables us to explore. Intuitively, we tend to take the action where $b$ is large in order to reduce the uncertainty and gradually attain accurate estimators. By adding a bonus $b$, we always maintain an **upper confidence bound** of the **exponentiated value functions**. This is true _even when we have only one sample_.

---

### Official Review · Reviewer_7Bjf · 2021-07-20

**Rating:** 6
**Confidence:** 4

**Summary:**

This paper proposes an algorithm and regret analysis of risk sensitive reinforcement learning. Their regret guarantee is significantly tighter than prior work; in particular, it helps close the gap compared to the lower bound.

**Limitations And Societal Impact:**

Yes.

**Main Review:**

Strengths
- Important problem
- Significantly tighter regret analysis compared to prior work

Weaknesses
- Unclear connection to distributional reinforcement learning
- Unclear exactly what novelty is being claimed in terms of connection to entropic risk measure

Overall, I think this paper is studying an interesting and important problem, and includes several new insights that allow them to improve upon prior work.

My main concern with this paper involves some significant issues with its exposition, in particular, regarding the claimed connection to distribution reinforcement learning. First, the connection to distributional reinforcement learning is itself unclear. As far as I understand, the connection is simply that the exponential value function is the moment generating function of the distributional value function. However, this connection seems to be completely irrelevant to the remainder of the paper. They also do not make this connection very precise, and it is also unclear to me whether this connection is novel.

The authors also claim in their contributions that the connection is in terms of the “entropic risk measure”; I didn’t understand what the authors meant here.


**Time Spent Reviewing:**

1

---

> ### Author Response · Authors · 2021-08-10
> **Response to Reviewer 7Bjf**
>
> Thank you for your positive comments. We would like to address your concerns as follows.
>
> - “connection to distribution reinforcement learning”
>
>    In distributional RL, one aims to estimate the value distribution, or in other words, the distribution of cumulative rewards. Inspired by the exponential Bellman equation (4), we may estimate the MGF of cumulative rewards. Note that an MGF uniquely identifies a distribution, just like quantiles or distribution functions used in existing distributional RL algorithms. To that end, in theory, one may obtain an estimate of the MGF of the reward distribution by running our Algorithm 1 or 2 for all $\beta \ne 0$. In practice, one may run the algorithms for values of $\beta$ within a predetermined grid. The choice of grid depends on the available computational power, prior knowledge or estimate of the heavy-tailedness of the reward distribution, and etc. To the best of our knowledge, this is the first time that a connection is made between distributional RL and risk-sensitive RL with entropic risk measure. We will add this clarification to the final version.
>
>
>
>
>
>
> - “novelty in terms of connection to entropic risk measure”
>
>    For risk-sensitive RL with the entropic (i.e., log-exponential) risk measure, our paper derives an improved regret bound compared to the bound in [16]. The improved bound is enabled by leveraging the exponential Bellman equation in the regret analysis and using a doubly decaying bonus mechanism in our algorithm design. To the best of our knowledge, both aspects are novel in the RL literature. Please see Line 49-58 in our paper for more details.

---

### Decision · Program_Chairs · 2021-09-27

**Decision:**

Accept (Poster)

**Comment:**

Reviewers believe that the improved regret bound in this paper makes a significant contribution and that the paper should be considered for acceptance on the basis of that contribution alone.

However, the one sticking point during reviewer discussion concerned the distributional RL connections in the paper.  More precisely, a long nested discussion starting with nUcE's initial review concerns about the mostly undeveloped link to Distributional RL (also reflected in other reviews) and continued towards the end of the discussion chain with reviewer tvfS.  The authors presented a detailed mathematical presentation in their final nested comment that convinced reviewer tvfS of the connection being claimed.

While ultimately the reviewer concerns were addressed with this final author response, it is critically important that this mathematical presentation and discussion be included on revision, perhaps in an Appendix.  Furthermore, while reviewer tvfS is satisfied with the author response, reviewer nUcE would still prefer that a revised paper focus on the regret bound contributions and focus on the distributional RL discussion as a "a side remark that opens to future interesting research directions".  The authors are asked to carefully consider all of these comments as they prepare their final version.